



# Two millennia of Main region (southern Germany) hydroclimate variability

Alexander Land[1], Sabine Remmele[1,2], Jutta Hofmann[3], Daniel Reichle[1], Margaret Eppli[1], Christian Zang[4], Allan Buras[5], Sebastian Hein[2], Reiner Zimmermann[1]

[1]University of Hohenheim, Institute of Botany (210a), Garbenstraße 30, 70599 Stuttgart, Germany
[2]University of Applied Forest Sciences, Schadenweilerhof, 72108 Rottenburg am Neckar, Germany
[3]Jahrringlabor Hofmann, Waldhäuser Str. 12, 72622 Nürtingen, Germany
[4]Technical University of Munich, Land Surface-Atmosphere Interactions, Hans-Carl-von-Carlowitz-Platz 2, 85354 Freising, Germany
[5]Technical University of Munich, Ecoclimatology, Hans-Carl-von-Carlowitz-Platz 2, 85354 Freising, Germany

*Correspondence to*: Alexander Land (alexander.land@uni-hohenheim.de)

**Abstract.** A reconstruction of hydroclimate with an annual or sub-annual resolution covering the entire Holocene for a geographically limited region would significantly improve our knowledge of past climate dynamics, but has not been developed so far. With the use of an extensive collection of oak total ring-width series (*Quercus robur* and *Quercus petraea*) from living trees, historic timbers and subfossil alluvial wood deposits from the Main River region in southern Germany, a regional, 2,000-year long, seasonally-resolved hydroclimate reconstruction for the Main region has been developed. Climate-growth response analysis has been performed with daily climate records from AD 1900 onwards. An innovative analysis method for testing the stability of the developed transfer function (bootstrapped transfer function stability test, BTFS) as well as a classical calibration/verification approach have been implemented to study climate-growth model performance. Living oak trees from the Main River region show a significant sensitivity to precipitation sum from February 26 to July 06 (spring to mid-summer) during the full (r = 0.49, p < 0.01, N = 116) and split (r = 0.58, p < 0.01, N = 58) calibration periods. BTFS confirmed the stability of the developed transfer function. The developed precipitation reconstruction reveals high variability on a high- to mid-frequency scale during the past two millennia. Very dry spring to mid-summer seasons lasting multiple years appeared in the decades AD 500/510s, 940s, 1170s, 1390s and 1160s. At the end of the AD 330s, a persistent multi-year drought with drastically reduced rainfall (w.r.t. 1901–2000) could be identified, which was the driest decade over the past 2,000 years in this region. In the AD 550s, 1050s, 1310s and 1480s, multi-year periods with high rainfall hit the Main region. In the spring to mid-summer of AD 338, precipitation was reduced by 38 % and in AD 357 it increased by 39 %. The presented hydroclimate reconstruction and its comparison to other records reveal interesting insights into the hydroclimate dynamics of the geographically limited area over the Common Era, as well as revealing noticeable temporal differences.





# 1 Introduction

The observed change in climate in recent decades has already impacted natural and human systems (Stocker et al., 2013). To predict future impacts on the earth system, it is necessary to investigate historical climate. Such investigations rely on various climate proxies which preserve past physical characteristics. Tree rings are widely used as a proxy to reconstruct past

climate variability and to provide information on climate fluctuations on a sub-annual basis. Changes in air temperature have been intensively investigated with temperature-sensitive tree-ring chronologies (here we refer to Wilson et al., 2016; Anchukaitis et al., 2017 and references therein) on a regional to global scale, while studies of long-term hydroclimate variability, rainfall and drought intensity (Briffa et al., 2009; Hughes and Brown, 1992; Esper et al., 2007; Cook et al., 2007; Stockton and Meko, 1975) are much more rare (Edvardsson et al., 2016). For Europe and some of its continental islands (e.g.

Great Britain), several hydroclimate reconstructions based on tree rings are available. They cover time periods ranging from centuries to, in a few cases, millennia and allow insights into sub-annual hydroclimate variability (Cooper et al., 2013; Wilson et al., 2005; Wilson et al., 2013; Levanič et al., 2013; Ruiz-Labourdette et al., 2014; Pauling et al., 2006; Cook et al., 2015; Seftigen et al., 2017). However, due to a lack of samples from ancient (subfossil) wood before the Middle Ages, there are far less hydroclimate reconstructions developed from tree rings before that epoch. Nevertheless, some studies addressed

this challenge and yield the unique possibility to investigate hydroclimate fluctuations beyond the Common Era (Land et al., 2015; Büntgen et al., 2011; Pechtl and Land, in press; Schönbein et al., 2015). For accurate climate model assessment, it is crucial to understand the full range of natural rainfall variability, which would allow to suit current models. To master this challenge, highly-resolved tree-ring reconstructions covering time spans as long as possible, at best over several millennia, which preserve fluctuations on short to long frequency scales are needed.

The historic instrumental data sets (~150 years) are too short to considerably reduce the uncertainty in hydroclimate projections (Ljungqvist et al., 2016). Thus, reconstructions of past hydroclimate variability from tree rings and from a distinct geographical region over substantial parts of the Holocene would allow us to study the natural (full range) dynamics on annual to decadal time scales. This would provide a baseline for climate model simulations and improve the verification of model outputs for predicting future droughts and pluvials.

In this study, a set of total ring width series from the Main region in southern Germany was used to achieve a robust reconstruction of spring to mid-summer precipitation variability. As hydroclimate is generally very localized, limiting the geographic extent of our study to the Main region allows us to create a model that can be used to infer rainfall variability over millennia. Additionally, we have chosen this region since there might be a high potential to develop a hydroclimate record covering the Early, Mid- and parts of the Late Holocene. An innovative bootstrapped transfer function stability test

(Buras et al., 2017) was used to assess the stability of the applied climate-growth model and the representativeness of the presented reconstruction. The connectivity to other existing hydroclimate reconstructions is shown and we critically discuss (a) the feasibility of developing a hydroclimate reconstruction spanning over substantial parts of the Holocene, (b) the



potential to obtain information on the frequency and intensity of severe droughts/wet periods lasting over seasons or even decades and (c) the independence of our tree-ring data set compared to other reconstructions from Central Europe.

## 2 Materials and methods

### 2.1 Tree-ring data

The Main region (hereafter referred to as MR) is located in Germany and gets its name from the Main River. The Main River originates in the Fichtel Mountains (northeast Bavaria, Germany) and after approximately 500 km empties into the Rhine River. The Main River is the fourth largest tributary of the Rhine and runs from east to west, which is rare for Central Europe. From the MR, total ring width (TRW) series were used to construct a composite oak TRW chronology covering the
period from AD 1–2015. To achieve this, an extensive set of 1,405 tree-ring series from this distinct geographical region is available. Such data are generally collected and stored in commercial and university dendro labs all over the world and used to develop robust TRW chronologies with sufficient replication. At the University of Hohenheim, for example, a tree-ring archive of ancient pine (Preboreal pine chronology, PPC) and oak samples (Holocene oak chronology, HOC) spanning the entire Holocene (~12,500 years, Friedrich et al., 2004) exists which serves as a suitable archive for studies of past climate
variability. The HOC consists of oak samples from quaternary deposits, archeological wood findings, construction timbers and living trees primarily sampled from southern Germany. Together, these form a unique, annually-resolved archive providing an excellent opportunity to study paleoclimate (for details about the PPC and HOC we refer to Friedrich et al., 2004).

The composite TRW chronology contains tree-ring series from living trees, construction timbers (historical) and alluvial
deposits (subfossil). TRW series from historical (Büntgen et al., 2010; Cooper et al., 2013; Wilson and Elling, 2004; Wilson et al., 2005) and archaeological/subfossil material (Kreuzwieser et al., 2004; Land et al., 2015; Schönbein et al., 2015; Pechtl and Land, in press) have already been used successfully in dendroclimatological studies to reconstruct past hydroclimate variability. All TRW series used for this study originated from a well-defined geographical region between 49.3–50.8° N and 8.2–12.2° E (Fig. 1).



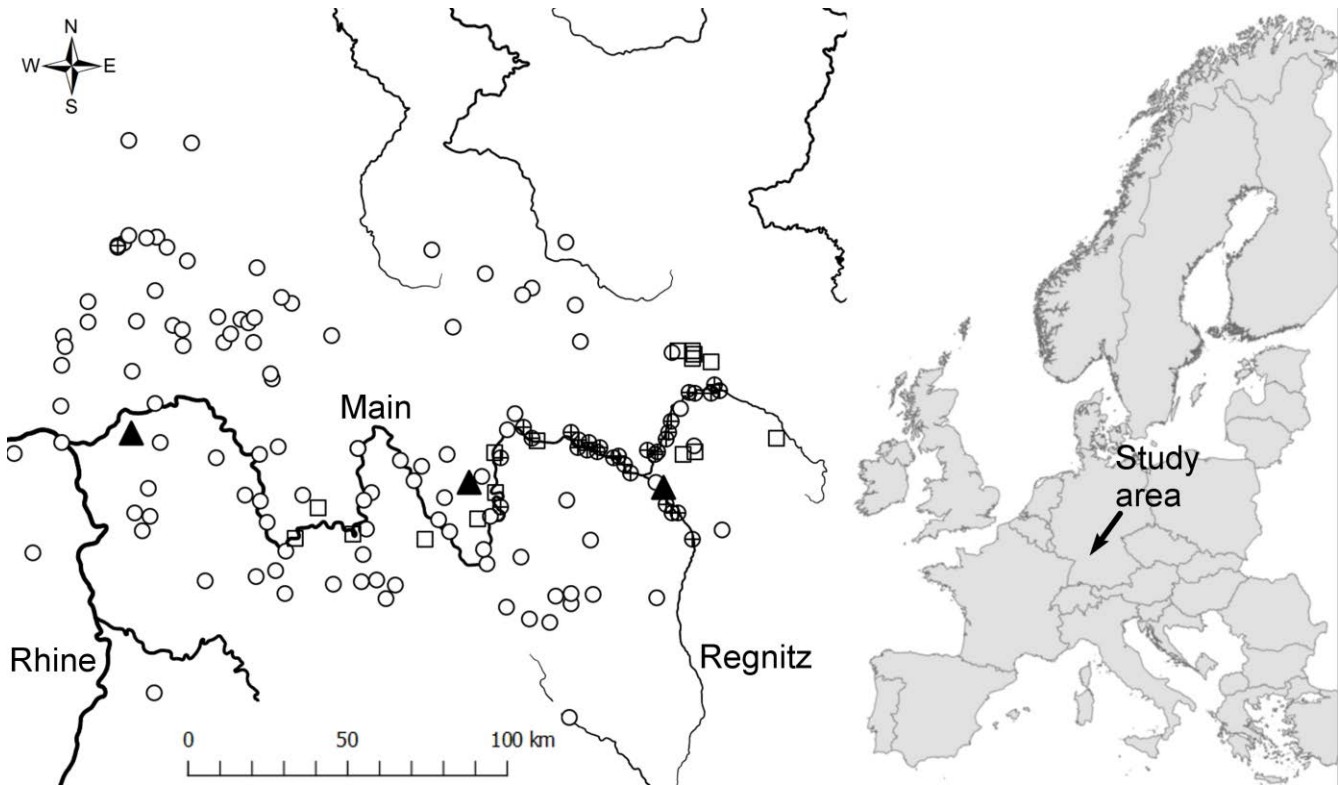

**Fig. 1. Study area.** Locations of the tree-ring sites and meteorological stations in the Main region. The study area is located between 49.3–50.8° N and 8.2–12.2° E. Sites of living trees (squares), localities of historical samples from construction wood (circles) and gravel pits of alluvial deposit subfossil oaks (circles with cross-hairs) are shown. The three meteorological stations (triangles) are located in the eastern, middle and western parts of the study area.

For calibration, living TRW series were used from locations along the middle and upper Main valley. The specific growth sites of construction timbers used in historical buildings are unknown, but provenance is assumed to be of local origin. The historical TRW samples are spread over the entire MR and were obtained from the tree-ring laboratory of the University of Hohenheim and the Hofmann tree-ring laboratory. The TRW series from subfossil oak trunks were derived from quaternary deposits of the upper Main River and the lower Regnitz River.

The entire data set of the original TRW series used for this study is available for public access (see Sect. Data availability).





## 2.2 Daily meteorological precipitation records

Daily precipitation sum (DPS) records from three long-term meteorological stations (Fig. 1) were used to assess the hydroclimate response of the TRW chronology. The meteorological stations Bamberg (49.88° N, 10.92° E, 240 m a.s.l.), Bergtheim (49.90° N, 10.07° E, 270 m a.s.l.) and Frankfurt a.M. (50.05° N, 08.60° E, 112 m a.s.l.) are located within the

5 MR. The stations were selected due to their long-term DPS records dating back to 1870 (Frankfurt a.M.), 1879 (Bamberg) and 1899 (Bergtheim) as well nearly complete status of these records. The data were provided and verified by KNMI Climate Explorer (http://climexp.knmi.nl., Klein Tank et al., 2002) and from the Deutscher Wetterdienst (DWD). The DPS record from the Bamberg station did not contain any missing data during the entire period, whereas the DPS records of the Frankfurt a.M. and Bergtheim stations had a gap between March and August 1945.

Precipitation sums of each day from all three stations from AD 1900 to 2015 were averaged to obtain a single DPS record representative for the MR. Between March and August 1945, only the Bamberg DPS record was used. This MR DPS was used for climate-growth analysis.

During AD 1901–2000 the mean precipitation sum between Feb. 26–Jul. 06 (spring to mid-summer) was calculated, serving as a reference period. This interval was chosen based on the results of the climate-growth analysis result during calibration.

## 2.3 Tree-ring series and chronology construction

For this study, precisely dated TRW series from oaks were assembled to construct a two millennia composite chronology (AD 1–2015) representing the seasonal radial growth variability within the MR.

The individual TRW series were primarily obtained from the Hohenheim tree-ring archive containing series from subfossil

(*Quercus* sp.), historical (*Q.* sp.) and living (*Q. robur*, *Q. petraea*) trees as well as from the Hofmann tree-ring laboratory (historical series, *Q.* sp.) to establish a well-replicated TRW chronology. Figure 2 shows the growth period, the mean segment length (MSL) as well as the running inter-series correlation (RBAR) and the Expressed Population Signal (EPS) of the TRW data set.





**Fig. 2. MR oak trees used for the reconstruction. (a)** Growth period of living, historical and subfossil tree-ring series, **(b)** mean segment length (MSL) and **(c)** inter-series correlation (RBAR) and Expressed Population Signal (EPS) statistics.

For the study, a set of 1,405 precisely dated tree-ring series (Küppers et al., 2018) consisting of 618 subfossil, 586 historical and 201 living trees was available (Fig. 2). The subfossil oaks cover the period until AD 1184 and some individual series go back to 321 before Common Era (BCE). Historical tree-ring series range from AD 797 to 1897 and the living series span the period from AD 1380 to 2015. High fluctuations in the replication throughout the study period can be observed, particularly

10  in the subfossil material, indicating a strong river dynamic which led to low / high deposition frequencies. A sufficient overlap between subfossil and historical as well as between historical and living TRW series was ensured.

The raw TRW series were detrended by fitting a fixed 100-year cubic smoothing spline (Cook and Peters, 1981) to each individual TRW series. A dimensionless index was obtained by calculating the ratio of the raw and predicted values. The



individual detrended TRW series were averaged using the robust bi-weight mean method to develop the standard TRW chronology. Due to fluctuations in the replication, variance stabilization was performed by applying the RBAR-weighted method (Frank et al., 2007; Briffa and Jones, 1990). The above-mentioned standardization procedure was conducted using the ARSTAN software (Cook and Krusic, 2005).

Expressed Population Signal (EPS) and inter-series correlation (RBAR) are key statistics in dendroclimatology for assessing the representativeness of a TRW chronology. EPS and RBAR statistics were calculated as a quality criterion of the coherence of the TRW chronology. Running EPS and RBAR were calculated during 100-year periods with a 50-year overlap. An EPS value of > 0.85 was defined as acceptable for a noise-free chronology. Even when concerns regarding the misinterpretation of the EPS threshold have recently been raised (Buras, 2017), we will use the above mentioned EPS

threshold to ensure that our study is comparable to others.

### 2.4 Calibration, verification and reconstruction of hydroclimate variability

Tree-ring climate response was assessed using a) a classical split-period calibration / verification (AD 1900–1957 / 1958–2015) and b) a full period calibration approach (AD 1900–2015). The correlation coefficient (r) was calculated for

calibration and the coefficient of efficiency (CE) for verification (Cook et al., 1994) to assess the reconstruction quality. A CE greater than zero was assumed to indicate a robust reconstruction.

To assess the temporal stability of the relationship between ring-widths and DPS, we applied the bootstrapped transfer function stability test (BTFS, Buras et al., 2017). BTFS bootstraps model parameter ratios between calibration and verification periods of equal length (here 58 years) for intercept, slope, and explained variance over 1000 iterations and tests

whether the obtained sample differs significantly from one which would indicate instability of the given parameter. Thus, p-values below 0.05 indicate unstable transfer function parameters.

Calibration was conducted between the TRW chronology of the living oak trees and the DPS record. Correlation analysis was performed using a MATLAB® (MathWorks, 1994-2008) script (Schönbein, 2011) which aggregated DPS data for each year, altering the length of the data interval (from 31 to 361 days in steps of ten days) and the date of start (between January

1 and December 15). Statistical significance was attained for α = 5 %. For details about the running of the script we refer to Schönbein (2011) and Land et al. (2017).

The developed linear climate-growth model was applied to the composite TRW chronology to reconstruct two millennia of hydroclimate variability for the MR. The statistical metric root mean square error (RMSE) was used to measure the climate-model performance.




## 2.5 Comparison to other hydroclimate reconstructions and independence of the data set

The developed reconstruction of the MR hydroclimate was compared to other hydroclimate reconstructions available for this region. Pauling et al. (2006) reconstructed the seasonal (spring, summer) precipitation from natural proxies (tree-ring chronologies, ice cores, corals and a speleothem) for European land areas from AD 1500 to 1900 (hereafter referred to as

P06sp, P06su). Cook et al. (2015) (hereafter referred to as C15) released a tree-ring based reconstruction of summer droughts and pluvials over Europe and Büntgen et al. (2011) (hereafter referred to as B11) reconstructed April–June precipitation sums over Central Europe, the latest two studies covering the past two millennia. The reconstruction series were obtained from the National Oceanic and Atmospheric Administration (http://www.noaa.gov) database (grid 49.75° N, 10.25° E, center).

The set of original TRW series used by the mentioned authors is, to the best of our knowledge, not stated in their work or accessible, which makes the comparison between the different records impossible. We were therefore unable to check for full independence between the different data sets, meaning that the data set used here might be not fully independent to others, particularly from Middle Ages to the beginning of the modern era. Nevertheless, for the periods in which subfossil and living TRW series have been used, a full independence of the MR exists to C15, B11 and P06. We assume that the

historical TRW series MR data set guaranty an independence of at least 55 % over the reconstruction period from AD ~1150 to 1700. At best, the majority (> 80 %) of the MR historical TRW series have not been used by C15, B11 or P06.

It should be mentioned that this is a sub-optimal situation which does not allow us to ensure the necessary independence between the data sets, but is unfortunately unavoidable in light of the data availability. However, the MR TRW data set used here gives the unique possibility to study the hydroclimate dynamic in a geographically limited area over two millennia.

To expose the common power and relative phase in time-frequency space between the aforementioned and the MR reconstruction, cross-wavelet transform (XWT) was evaluated and wavelet coherence (WTC) was measured by using a MATLAB® script from Grinsted et al. (2004). The XWT finds regions with high common power and WTC shows where the compared series co-vary in time-frequency space (even when common power is low). For all calculations, the Morlet

wavelet was chosen, providing a good balance between time and frequency localization (Grinsted et al., 2004). Additionally, a 51-year running correlation between the MR reconstruction and P06sp, P06su, C15, B11 was performed.





# 3 Results

## 3.1 TRW composite chronology

The number of TRW series per year fluctuates over time. The lowest replication can be found around AD 280 / AD 850, while the highest is in the first decade AD (Fig. 2a). Mean segment length (MSL) is longer for the subfossil and living TRW

5 series and shorter for the historical series, but always exceeds 100 years (Fig. 2b). Expressed Population Signal (EPS) is always above the threshold of 0.85 during the entire time span (mean EPS = 0.96). The lowest EPS was observed in the transition period of AD 830 where subfossil and historical TRW series overlap, corresponding with low replication. The mean inter-series correlation (RBAR) is 0.32 (Fig. 2c) showing low values in the first half of the 14th and at the end of the 17th century where the TRW chronology consists of primarily historical TRW series. In periods where only subfossil and

10 living TRW series were used, RBAR increases and is considerably higher, e.g. prior to AD 500 (subfossil) as well as after AD 1830 (living). Further worth mentioning, is that the subfossil TRW series seem to be much more homogenous in regards to their inter-series growth pattern than the historical series. The relatively small region of deposited subfossil trees (Fig. 1) could provide an explanation.

## 15 3.2 Climate-growth model and hydroclimate reconstruction

Figure 3 illustrates the calibration and verification process implemented for the development of the model that was used for further hydroclimate reconstruction.







**Fig. 3. Calibration and verification of the MR oak climate-response analysis. (a)** Full (AD 1900–2015) and **(b)** split (AD 1958–2015) period calibration. The model was verified during AD 1900–1957. Actual precipitation sum from Feb. 26–July 06 (blue) and reconstruction (black) are shown.

The TRW chronology reveals a significant relationship (p < 0.01) to DPS from Feb. 26–July 06 (spring to mid-summer season) during the full (r = 0.49) and split (r = 0.58) calibration period. A CE of 0.23 accounts for a robust reconstruction. This indicates that the developed climate-growth model is suitable for reconstructing regional hydroclimate variability. Nevertheless, as can be seen from Fig. 3, the TRW chronology does not track extremely low (e.g. 1903, 1976, 1991, 1993, 10  2003, 2015) or high (e.g. 1965, 2007) precipitation rates adequately. Thus, the model underestimates the true sum of precipitation in the spring to mid-summer season. Similar results were published by other authors (Cooper et al., 2013; Wilson et al., 2013). Based on the work of McCarroll et al. (2015), we simply defined a 10 % threshold of years in which precipitation sum was low / high during the full calibration period (116 years, 10 % = 11.6 years each) as extremes. After




ranking the precipitation data, we checked how many years of the TRW data capture these extremes. Only three years lie beyond the lowest (statistically not significant), but four years beyond the highest threshold (statistically significant, $p < 0.05$). Even though a close statistic relationship between DPS and TRW is evident, there is spurious correlation regarding extreme values.

The applied bootstrapped transfer function stability test (BTFS) indicated stability of transfer function parameters (slope, intercept, $r^2$) over time ($p \geq 0.25$ w.r.t. to the null hypothesis of perfectly stable model parameters). A highly significant sign-test ($p < 0.001$) revealed true collinearity between TRW and DPS over the calibration period. Model residuals were normally distributed and did not express significant autocorrelations. A moving window correlation between TRW and DPS revealed

10   significant temporal correlations over the calibration, however these varied between 0.33 ($p < 0.05$) and 0.60 ($p < 0.001$). Both classic calibration/verification as well as BTFS confirm the sensitivity of TRW to Feb. 26–July 06 precipitation totals.

The developed climate-growth model was applied to the composite chronology to reconstruct spring to mid-summer (Feb. 26–July 06) precipitation variability for the MR from AD 2015 back to AD 1. The seasonally-resolved reconstruction series

15   over two millennia is shown in Fig. 4. The reconstruction maintains high seasonal as well as decadal scale fluctuations.

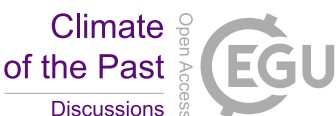



**Fig. 4. Reconstruction of spring to mid-summer season precipitation variability for the MR.** Reconstruction of precipitation sum from Feb. 26–July 06 (black) and 10-year lowpass filter (red) during AD 1–2000. RMSE = light grey shaded, thin white line = mean precipitation sum during the reference period AD 1901–2000.

The developed reconstruction shows high variability on an annual to decadal time scale with repeated phases of below / above average rainfall. According to our findings, nine years of less than 25 % of mean precipitation sum (w.r.t. AD 1901–2000) could be detected (Table 1). The years AD 338 and 337 are marked by severe low pluvials in the MR. In the decade

from AD 334 to 344, the MR was exposed to a conspicuous drought far below rainfall average and thus it can be described as an extraordinarily dry period. In the 12th century (around AD 1165 and 1177) and in the last decade of the 14th century (around AD 1395), three additional periods of below average rainfall occurred. Heavy single-season rainfall exceeding the mean precipitation sum by a minimum of 25 % (w.r.t. AD 1901–2000) appeared in 26 individual years during the past two

millennia (Table 1). The years AD 357 and 985 appear as the wettest seasons with high deviations from the reference period. Periods of unusually high rainfall lasting five years or longer appear in the mid-6th, mid-11th and at the beginning of the 14th century. The period from AD 1125 to 1138 is characterized by consistently high pluvials exceeding the average seasonal precipitation sum for almost one and a half decades.

It can be summarized that the Main region suffered from severe long-lasting droughts and pluvials over the past two millennia, especially during the mid-4th century.

### 3.3 Comparison of MR reconstruction to others

To evaluate the connection between the MR reconstruction and C15, B11 and P06sp/su, a running correlation (Fig. A1) as

well as wavelet analyses between MR and C15 / B11 were performed (Fig. 5).

The 51-year running correlation accounts for temporal variation between the reconstructions. The correlation between MR and C15 / B11 is high from the 12th until the 18th century. Before the 12th and from the 18th century onwards the connections declined to sporadic cases. Around AD 300 and 400 as well as during the 9th century, the MR shows only weak connections to C15 and B11. In the mid-19th century, MR and B11 hold no statistical relationship. When the MR is

compared to P06sp and P06su, a relationship is only evident during short time periods. Especially during the end of the 17th to the mid-18th century as well as over the 20th century, a connection to P06sp/su is not detectable. In general, MR and P06sp/su are only spuriously related.

Figure 5 shows the cross-wavelet transform (XWT) and the wavelet coherence (WTC) between MR-B11 and MR-C15 over

two millennia.





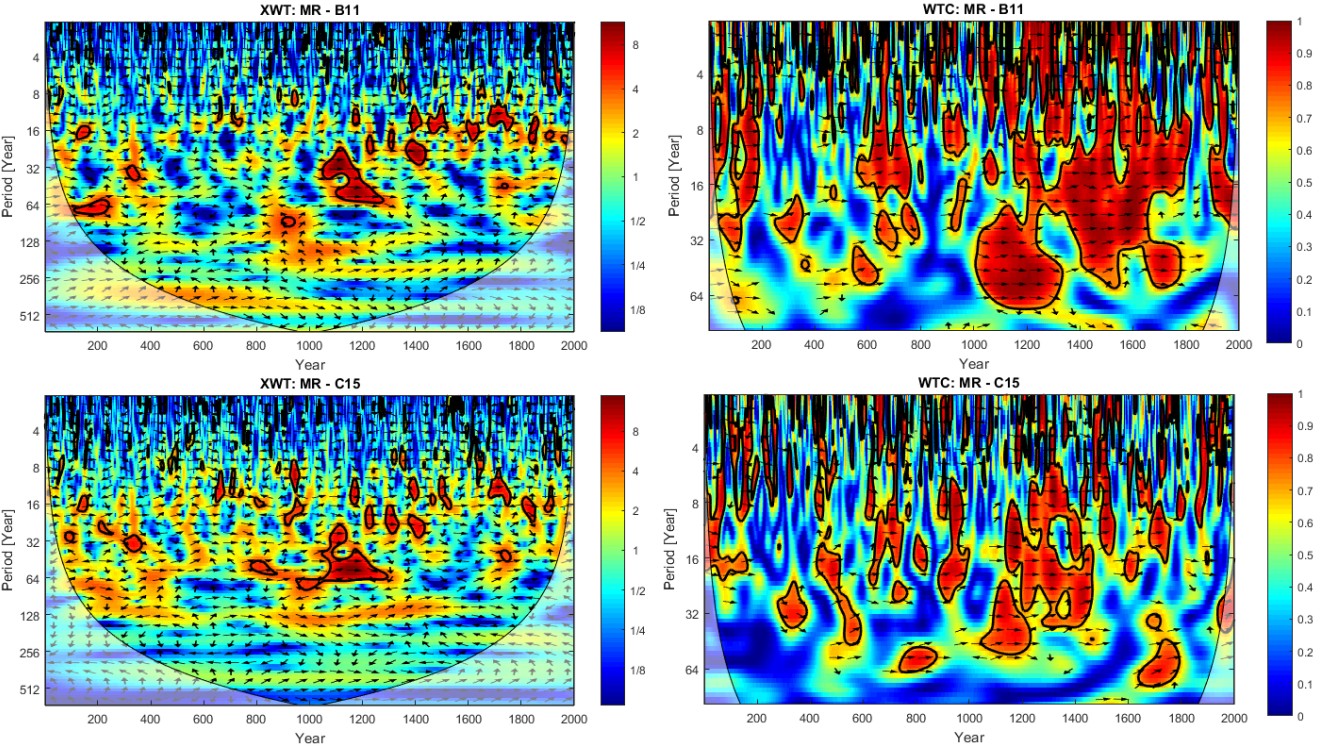

**Fig. 5. Cross-wavelet transform (XWT, left) and squared wavelet coherence (WTC, right) between MR-B11 (top) and MR-C15 (bottom) from AD 1–2000.** 5 % significance level against red noise is shown as a thick contour. Light shading depicts the cone of influence.

The XWT between MR-B11 shows significant common power in the ~10–15 year band from the 12th century onwards and in the ~20–60 year band around the 11th and 12th century. The XWT between MR-C15 depicts similar results. Albeit, the ~23–65 year band from AD 1000 to 1300 is more pronounced. Looking at the squared coherence spectrum (WTC) between MR-B11 and MR-C15, the close connections appear on the ~4 year band and are more pronounced on the 10–60 year band, especially after AD 1000. High coherence between MR-B11 and MR-C15 is also obvious prior to AD 1000, on short as well as on long terms. Nevertheless, there is a distinct difference between pre- and post-AD 1000.

Figure 6 shows the 20-year lowpass filtered reconstructions from MR, B11 and C15 from AD 1 to 2000 (Fig. 6a) as well as for two sub-periods with an annual resolution (Fig. 6b/c).



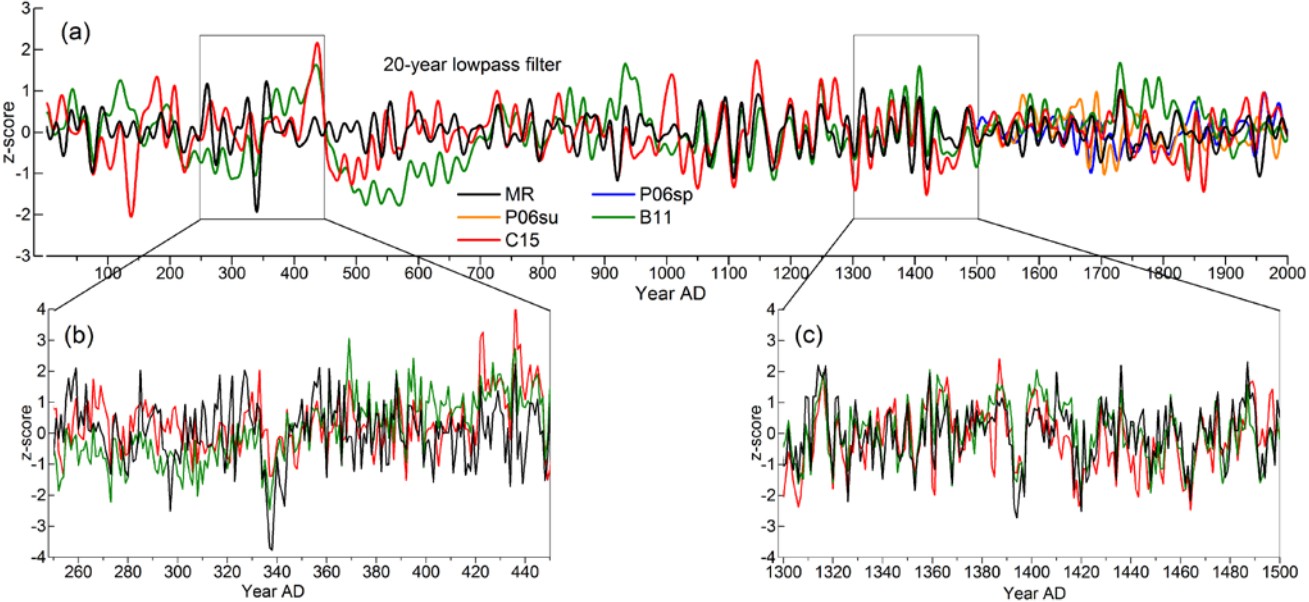

**Fig. 6. Smoothed hydroclimate reconstructions from AD 1–2000.** (a) 20-year lowpass filtered, (b) annual fluctuations from AD 250–450 and (c) from AD 1300–1500.

As shown in the previous section, the reconstruction series contain differences and similarities on year-to-year and on mid-frequency fluctuations. It is obvious from Fig. 6a that MR, C15 and B11 hold strong discrepancies regarding their fluctuations until the end of the 7th century (see also Fig. 5). For example, B11 shows a much higher fluctuation on a longer time scale between AD 250 and 700 than C15 and MR. However, during the aforementioned dry decade around AD 338 (Fig. 6b), all three reconstructions independently show similar patterns and document a long-lasting drought. On the other

hand, the pronounced long-lasting drought in the 6th century as reconstructed by B11 is not seen in MR or C15. Although C15 and B11 show evidence of a severe wet decade around AD 435, the MR does not confirm this result. On the contrary, the MR shows moderate changes on a mid-frequency scale. From the end of the 11th to the mid-18th century, all reconstruction series show highly similar patterns (Fig. 5; Fig. 6c; Fig. A1) from year-to-year and on mid-frequency, with the exception of P06sp/su. They share changes in the amplitude synchronously even in very wet and severe dry years/periods. In

particular from AD 1800 onwards, when full independence of the MR data set is guaranteed, notable differences occur. Running correlation analyses also account for substantial differences between the MR and the other reconstructions over the last two centuries (see also Fig. A1).



## 4 Discussion

### 4.1 Climate-sensitivity of the MR TRW chronology and stability of transfer function

The oak TRW chronology from the Main region (MR) in southern Germany reveals a significant ($p < 0.01$) sensitivity to the precipitation sum from February 26 to July 06 (spring to mid-summer season). During calibration and verification it became

clear that TRW does not cover the full range of precipitation variability, resulting in a poor correlation between TRW and extremes. Low and high seasonal precipitation sums are not accurately reflected in the applied climate-growth model, which is also mentioned by e.g. Cooper et al. (2013) and Wilson et al. (2013). Land et al. (2017) pointed out that oak tree-ring series are not sensitive to short, heavy rainfall events during the growing season, which may provide a further explanation for this fact. Another reason could be a loss of photosynthetically active leaf area caused by fungal infestations (e.g. mildew)

during such phases of high precipitation. We further note that in the MR, a massive insect attack took place between AD 1954–1958 (Steger, 1959, 1960) which reduced growth considerably and certainly influenced the climate-growth model performance. The complexity of oak growth over the course of the 20th century is also well-established. Friedrichs et al. (2008) showed that oak growth can increase when a combination of warm and dry conditions occur. This leads to the assumption that the control of oak growth remains complex, at least during isolated periods. Altogether, the applied climate-

growth model considerably underestimates the total seasonal precipitation sum which leads to a loss of explained variance.
The precipitation sensitivity of the MR oaks does, however, agree with findings from other studies conducted with oak tree-ring series in Europe. Büntgen et al. (2011) reported a sensitivity of the seasonal precipitation sum from April to June (Central Europe), while Friedrichs et al. (2008) (central-west Germany) and Čufar et al. (2008) (southeast Slovenia) for the June rainfall sum. Cooper et al. (2013) (East Anglian), Wilson et al. (2013) (south-central England) and Karanitsch-Ackerl et

al. (2017) (northeast Austria) found a close relationship to March–July precipitation sum and Land (2014), Land et al. (2015), Land et al. (2017) and Schönbein et al. (2015) (Franconia, Germany) to spring-summer precipitation.
European oak TRW chronologies show an explicit connection to hydroclimate during the growing season, in particular from spring until mid-summer.

### 4.2 Main region hydroclimate dynamic

The developed climate-growth model leaves a high level of unexplained variance, which is most apparent in seasons with far below / above average rainfall totals. This leads to an inevitable underestimation of past climate variability and is also reported from other authors (Cooper et al., 2013; Esper et al., 2005; Storch et al., 2004; Wilson et al., 2013). Thus, we can assume that in the MR, the year-to-year dynamic might be much more variable than suggested by the tree rings due to the

tree rings' inability to capture years with extreme low / high precipitation. This would mean that during the past two millennia, very low / high pluvials are much more pronounced and thus the hydroclimate dynamic is stronger than suggested by the presented reconstruction. On the other hand, a severely reduced growth of oak trees does not necessarily mean a



"true" dry season, but may instead be due to a combination of moderate rainfall, a warm spring season and / or an insect attack. Oak trees are prone to insect attacks, especially under natural forest dynamic processes. Capturing non-climate-driven tree-ring fluctuations, e.g. caused by insects, would therefore necessitate an investigation of wood anatomy which, due to the high number of wood samples, would be labor-intensive and cost-prohibitive. An attempt of such can be found in Land et al.

(2015) and Schönbein et al. (2015). Nevertheless, single seasons with pronounced below (e.g. AD 338) / above (e.g. AD 357) average rainfall totals as well as very dry several-year periods (e.g. around AD 1395) appeared in the MR. Interestingly, Spurk et al. (2002) investigated the depositional frequency of subfossil oaks in the MR, which includes the tree samples used here, and linked them to climatically-induced fluctuations. They found a sudden onset of germination at AD 400 in the MR, indicating humid conditions. From AD 400 onwards, our reconstruction gives no evidence for a continuous long-lasting

period with above-average rainfall, but shows distinct high year-to-year fluctuations between AD 420 and 550. Despite that, changes on a low frequency scale over centuries to millennia are masked by the standardization procedure.

As mentioned by Spurk et al. (2002), human influence could have had a severe impact on forest structure and forest dynamic since the third millennium before Common Era and may have therefore impacted the dynamic of tree growth, perhaps leading to a bias in the established reconstruction. Therefore, investigating epochs where human impact is low, or at best did

not occur, is crucial to get a clear picture of pre-human time and its climate dynamic. However, it should be mentioned that humans already settled in the MR over 7,000 years ago (Bickle and Whittle, 2013) and certainly influenced their environment accordingly. Much more effort must be made in the future to fully understand the hydroclimate dynamic on a small scale (e.g. in the Main region) during the entire Holocene. Even when the tree rings used represent a highly-resolved natural archive, combining different scientific fields, e.g. as recently done by Pechtl and Land (in press), is very necessary to

capture potential human effects on former forests. Another factor plays a crucial role: for southern Germany (and for large parts of Central Europe) oak is the only species from which TRW series are available from present back to the Early / Mid-Holocene. It is therefore of tremendous interest for paleoclimate research to interpret the growth fluctuations more precisely and to enhance climate-growth model performance.

**4.3 Comparison and independence of MR to other hydroclimate reconstructions**

The reconstruction of spring to mid-summer precipitation variability for the MR developed here shows high connectivity to C15 (Cook et al., 2015) and B11 (Büntgen et al., 2011) for substantial parts of the past two millennia. Nevertheless, the results of running correlations and wavelet coherence analyses clearly speak for notable differences during some periods. These differences are particularly obvious around AD 300–400, 750–850 and 1820–1870, accounting for intervals where the

MR TRW data set is fully independent from others. The differences can primarily be observed on a mid-frequency scale but are also seen in year-to-year fluctuations. We assume that one reason for this could lie in the differing standardization procedures utilized in chronology construction, but the use of mixed TRW series from various provenance and from different sources (archaeological findings) is also conceivable (for more detail we refer to the mentioned studies).

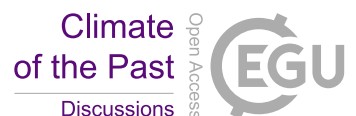

The high connectivity between MR-C15 and MR-B11 over ~700 years (end of 11th to mid-18th century) is outstanding and supports the assumption that duplicates within the TRW data are used in the different studies. The exact level of dependency between the different data sets cannot be stated here and remains undetermined. Thus, we assume that during this period our TRW data set might be not as independent as initially considered, which once again highlights the importance of data transparency.

Nevertheless, the MR hydroclimate reconstruction accounts for an on-site variability.

Developing a hydroclimate reconstruction for Central Europe reaching back to the Early / Mid-Holocene requires a large data set with a sufficient number of TRW series. This goal could be achieved with the use of TRW series from the Holocene oak chronology (HOC) Hohenheim (Friedrich et al., 2004), assuming that these subfossil oaks reflect the representative Central European hydroclimate. The results of our study support the assumption that the TRW series from the Main region, consisting of a well-replicated TRW data set (for more details see Friedrich et al., 2004; Leuschner et al., 2002; Spurk et al., 2002), have the potential to reflect the rainfall variability of Central Europe, at least for southern Germany, as well as regional hydro-regime aspects. Nevertheless, a ten millennia hydroclimate reconstruction would require the inclusion of TRW data from other river systems (e.g. upper Danube, upper Rhine) due to changes in the deposition frequency of oak trunks and the resultant variance in replication (Spurk et al., 2002; Friedrich et al., 2004). Such temporal changes of distribution are known from tree-ring archives of Ireland, northern Germany and the Netherlands (Leuschner et al., 2002; Spurk et al., 2002) as well. Combining such TRW data sets lasting for millennia from different regions or countries across Europe is crucial for the development of a well-replicated reconstruction representing the overall European seasonal rainfall variability during substantial parts of the Holocene. Additional regional investigations of hydroclimate variability that cover multiple millennia must also be conducted.

## 5 Conclusions

We conclude that oak TRW in the Main region is suitable to reconstruct past hydroclimate conditions with a seasonal resolution during the past two millennia as well as for the investigation of rainfall intensity on a high- to mid-frequency scale. Moreover, the developed hydroclimate reconstruction is, to the best of our knowledge, the only record covering two millennia with seasonal resolution for a geographically limited area in Central Europe. When the presented hydroclimate reconstruction is compared to others, it becomes apparent that significant differences on an annual to decadal scale appear under the assumption of a fully independent TRW data set. This raises the question about the potential to upscale or to transfer these results to a wider geographical extent. Finally, we hypothesize that a tree-ring hydroclimate reconstruction over the entire Holocene is feasible using a large TRW data set from different river systems in southern Germany, which also may hold the unique potential to reveal insights into Central European hydroclimate fluctuations on a seasonal scale.





**Data availability**

The entire data set of the original TRW series can be accessed: https://zenodo.org/deposit/1453330 (doi: 10.5281/zenodo.1453330).





# Appendices

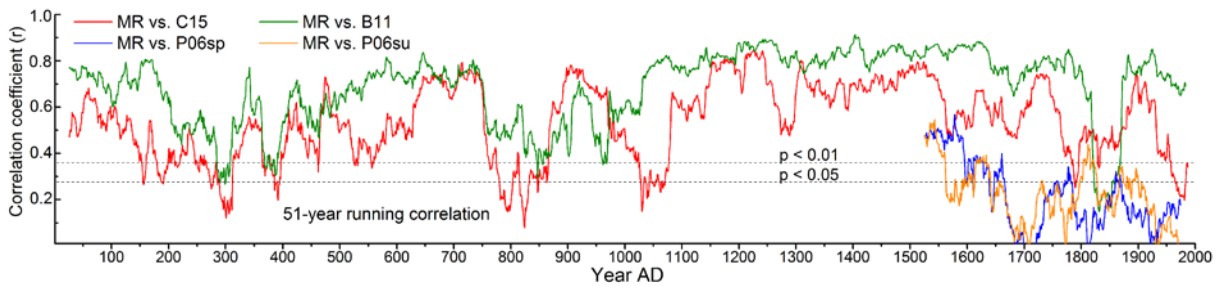

**Fig. A1. Comparison of MR precipitation reconstruction to other reconstructions of rainfall variability.** Statistical comparison (51-year running correlation) between the MR reconstruction and C15 (Cook et al., 2015), B11 (Büntgen et al., 2011), P06sp/su (Pauling et al., 2006).



**Author contribution**

AL compiled the dendroclimatic analyses, the hydroclimate reconstruction and wrote the manuscript. SR, DR, ME and SH processed the TRW measurements and wrote parts of the manuscript. CZ and AB performed the bootstrapped stability test and wrote parts of the manuscript. JH particularly provided historical TRW data. All authors read and approved the final
5   manuscript.

**Competing interests**

The authors declare that they have no conflict of interest.

**Acknowledgments**

We are grateful to Bernd Becker, Marco Spurk and Michael Friedrich for continuous sampling of trees from alluvial deposits
10   and construction timbers. We especially thank Bernd Becker, as he set up the Holocene oak chronology (HOC) Hohenheim with great personal effort. We acknowledge Manfred Küppers for critical notes on the study and for the provided data set of the HOC. This study greatly benefited from comments of an anonymous reviewer on a previous version of the manuscript.

**Funding**

This research did not receive any specific grant from funding agencies in the public, commercial, or not-for-profit sectors.





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





**Tables**

**Table 1. List of years with reconstructed far below /above pluvials (Feb. 26–Jul. 06) depicted as deviation (%) from the reference period AD 1901–2000.**

| Year AD | Low pluvials | Year AD | High pluvials |
|---------|--------------|---------|---------------|
| 338 | -38 | 357 | 39 |
| 337 | -37 | 985 | 37 |
| 1167 | -29 | 526 | 34 |
| 510 | -28 | 1533 | 34 |
| 1394 | -27 | 654 | 32 |
| 565 | -26 | 1317 | 29 |
| 945 | -26 | 1436 | 29 |
| 1165 | -26 | 460 | 29 |
| 1177 | -25 | 1314 | 28 |
| | | 436 | 28 |
| | | 559 | 28 |
| | | 1673 | 27 |
| | | 1123 | 27 |
| | | 1487 | 27 |
| | | 43 | 27 |
| | | 1052 | 26 |
| | | 1531 | 26 |
| | | 496 | 26 |
| | | 1055 | 26 |
| | | 56 | 26 |
| | | 778 | 26 |
| | | 1727 | 26 |
| | | 102 | 25 |
| | | 1056 | 25 |
| | | 932 | 25 |
| | | 602 | 25 |