# Peer review of "Two millennia of Main region (southern Germany) hydroclimate variability"

_Climate of the Past, 2018_

## Referee Comment (RC1) · Anonymous Referee #1 · 19 Nov 2018

Dear authors, I very much appreciate the attempt of analyzing and publishing some (why not all) of the unique oak ring-width data stored at the University of Hohenheim, Germany. This is a very important step towards generating exciting new science. However, I am reluctant in recommending acceptance of the submitted work, mainly because it represents an intermediate step rather than drawing methodological-sound conclusions from a final dataset (i.e. entire Holocene). This stepwise publishing procedure seems unnecessary in the case of central European oak ring-width measurements, as it has been successfully demonstrated in the recent past that such data are useful (and most relevant indeed) for reconstructing hydroclimate (i.e. a more complex reflection of spring to early-summer soil moisture availability instead of simple precipitation totals) on inter-annual to multi-centennial time-scales. While this has been done

for several regions in Germany, France and England, Ed Cook's OWDA describes a European-wide milestone with regard to spatially explicit reconstructions of droughts and pluvials during the Common Era. In addition, the authors suggest that some of their data has been already incorporated into earlier studies (Büntgen et al., 2011, Cook et al., 2015), therefore only limited comparisons with these reconstructions are possible. Why did not the authors clarify this before? I am confident Büntgen and Cook would provide this information to the authors. In short, the submitted work does not provide any ground-breaking methodological and/or intellectual novelty, and the relatively small data(sub)set does not appear to be robust between ~800 and 1100 C.E. and again during the 4th century C.E. when the sample size dramatically drops (see Fig. 2 of the submitted draft). Although the EPS is above the common applied threshold of 0.85, the temporal replication changes can strongly affect the chronology behavior. Possible uncertainties might emerge from the integration of predominantly juvenile or mature/adult wood during these periods. Moreover, it is a pity that the low-frequency hydroclimate variability is not expressed in the presented reconstruction.

---

## Author Comment (AC1) · 21 Nov 2018

Dear Referee #1,

Thank you for your comments on our work.

In your review you raised some critical points:

RC#1: Stepwise publishing of data

I think there is a misunderstanding here. Ring-width data stored at the University of Hohenheim (Germany) are regularly contributed to different projects/studies to generate exciting science. Tree-ring data from the dendrolab Hohenheim are contributed to a wide variety of different dendro projects (only a few examples are given): 1) "Five mil-

lennia of European hydroclimate" (head of the project: W. Tegel, University of Freiburg, Germany; U. Büntgen, University of Cambridge, UK), 2) "Long-term trends in European tree growth over the past 1000 years - an interspecies comparison" (head of the project: A. Seim, University of Freiburg, Germany, funded by the DFG, project no. 389131207). Most of the Hohenheim ring-width data are already provided via data repositories (e.g. PANGAEA). Smaller data(sub)sets are indeed not published yet and are only accessible via personal contact/correspondence. This is mainly ring-width data from very local findings or sites (e.g. from the Rhine river) spanning a few centuries within the Holocene, but NOT the entire Holocene. For the presented study, tree-ring series (you called that a data(sub)set) from a specific region (here the Main region, southern Germany) were needed to develop a regional hydroclimate record and to compare this record to already existing reconstructions (during the past two millennia). The original (raw) dataset used here is made accessible to give others the opportunity to specifically reproduce our results and to have unrestricted access to all data underlying our study. This is transparent, in line with the data policy of the Copernicus Publications and good practical science. This is not in any way "politically" motivated or "strategically" aligned. Any suspicions that we have deliberately withheld tree-ring data are completely unfounded. We, as a research group, have strived in recent years to make our institution a positive example of transparency and scientific cooperation. We are more than happy to collaborate with and provide our data to others in the hopes of gaining new insights into past climate.

RC#1: Reconstruction of hydroclimate is a more complex reflection of soil moisture availability instead of simple precipitation totals

The growth of tress is influenced by a couple of different biotic/abiotic factors. Intensive climate-growth analyzes were performed to develop a climate-growth model. In addition to precipitation totals, we used 22 indices (scPDSI, SPI, SPEI etc.) to evaluate the sensitivity of the tree-ring chronology to these indices. The record of "simple" precipitation totals performed best, and as we mentioned in chapter 3.2 (Climate-growth

model and hydroclimate reconstruction, Fig. 3) in years with very low/high rainfall the chronology does not track these extremes satisfactorily. It could be that in these years other factors influenced the growth of the oak trees more intensively than in other years. However, I propose that the "simple" precipitation totals give the best result we can get from these data.

RC#1: Limited comparison to other reconstructions

I am pretty sure, and so I do agree with your comment, that if requested, Büntgen and Cook would have provided their datasets for a detailed analysis regarding duplicates in the dataset used here, which would allow for a clear statement of independence/dependence between the different reconstructions. Perhaps I am totally wrong and the original datasets are accessible via a data repository, or there were good reasons not to make these datasets accessible/public for unknown reasons to me. The widely accepted data policy of scientific journals requires all authors to make materials, data etc. available. So one could argue that an independent study should be feasible (even without a direct correspondence during the publication process). I would like to mention that the mistake in this regard could be that I have simply failed to find the data (and the original datasets are accessible/public), meaning that this part of the manuscript has to be reanalyzed and modified.

RC#1: The work does not provide ground-breaking (methodological/intellectual) novelty

In this study we used precipitation records with a daily resolution, which is relatively new to dendroclimatology (chapter 2.4 Calibration, verification and reconstruction of hydroclimate variability). The applied bootstrapped transfer function stability (BTFS) test to assess the temporal stability of the relationship between ring-widths and daily precipitation data (first introduced by Buras et al. 2017) is another example for a new (and innovative) method that was used. While I do agree that most of the presented results were obtained using standard and widely accepted dendroclimatological methods, the presented results nevertheless show e.g. that in the first millennium C.E. (fully-independent dataset) differences to other reconstructions (B11, C15) appear, which could be due to local/regional precipitation characteristics (see chapter 3.3 Comparison of MR reconstruction to others, Fig. 5, Fig. 6). This underlines the need to set up as much as possible local/regional hydroclimate reconstructions (even when standard methods are applied) to study spatial and temporal rainfall variability in the near future. Thus, this work does indeed provide additional information leading to a more detailed understanding of climate variability (I also refer to the following comment).

RC#1: The dataset is not robust (4th century and from 800-1100 C.E.)

In the past few years intensive sampling of subfossil trees in the Main region was conducted, but did not lead to an increase in sample size in the 4th century and from 800-1100 C.E. The drop in sample size (as well as in the mean segment length) in the mentioned periods give evidence for fundamental environmental changes in the Main valley. It is possible that uncertainties in the reconstruction could emerge from the use of predominantly juvenile trees in these periods, but it underlines the statement (see previous comment) that there is a strong need to develop hydroclimate reconstructions on local/regional scale with tree-ring width datasets to ensure whether the uncertainty occurred from e.g. changing sample size.

―――――――――――――――――

---

## Referee Comment (RC2) · Anonymous Referee #2 · 12 Dec 2018

Dear editor and authors of the manuscript "Two millennia of Main region (southern Germany) hydroclimate variability". To the best of my knowledge, the 2000 years long chronology is a novel idea through integration all available tree-ring width samples in this work, which would be an important contribution in the denderochronology community. Another new information is to calibrate the tree-ring width chronology using the daily instrumental data. However, the robust of the reconstruction should be furtherly analyzed, and the mechanism of precipitation variability should be conducted for reader to understand the origin of variability in the high-impart journal Climate of the past. Thus, I suggest that the manuscript should be accepted for publication after a revision. Main comments: 1. There are very long chronologies in Europe where is a hotspot in dendroclimatology. It is highly encouraged to carefully review the previous

studies to place much more stress on innovation or difference of this study. The current motivation of this study is not very attractive to me. e.g. the first sentence in the abstract, the climate reconstruction covering the entire Holocene is important, but the TRW chronology in this study only covers the past two millennia. 2. The mechanism and origin of the precipitation variability (e.g. the influence of the Northern Atlantic Oscillation) should to be furtherly analyzed through comparison of the other reconstructions or model simulation. Another option is to select a more specific journal, e.g. dendrochronologia or Tree-Ring Research. The readers of Climate of the past would like to know some information about the mechanism of climate variability not only the phenomena. 3. The logic of the article is a bit problematic. The main target in this study is to reconstruct the precipitation variability over the past millennia. However, the following some evidences and discussion do not support this reconstruction. e.g. Page 10, line 9 'The TRW chronology does not track extremely low or high precipitation rates adequately.' The sections 4.3 also shows this weak relationship between the TRW chronology and the instrumental precipitation. Even the authors emphasize that the human influence may have a severe impact on forest in Page 17, lines 12-20. All prove that it is not very reasonable to reflect the extreme precipitation events over the past millennia. Another option to try to reconstruct the 'mixed' variable, e.g. the PDSI index. Specific Comments: 1. Page 1, Lines 18-19. The bootstrap method is not an innovative analysis in dendroclimatology, please see the literatures e.g. (Guiot 1991; Till and Guiot 1990). 2. Page 2, Line 3. It is difficult to predict future impact through climate reconstruction. The climate model is usually used to project the future scenario. 3. Page 6, Line 13. Why is the 100-year cubic smoothing spline used to detrend the tree growth? To my knowledge, the standard standardization method depends on the special situation of each sample. 4. Page 8. The section 2.5 should be moved to the introduction to emphasize the innovation of this paper through a review of previous reconstructions. 5. Page 8, Line 26. The reason selecting 51-year should be given. As we known, the window size would affect the results of running correlation. 6. The discussion of phase variability in the cross-wavelet transform and squared wavelet coherence is ignored. e.g. The MR and B11 has an obvious variability in phase in the upper right panel of Figure 5. 7. Page 18, Line 24. The seasonal resolution would lead a misunderstanding. Here, it is really an annual resolution.

References: Guiot J (1991) The bootstrapped response function. Tree-ring bulletin Till C, Guiot J (1990) Reconstruction of Precipitation in Morocco Since 1100 A.D. Based on Cedrus Atlantica Tree-Ring Widths. Quat Res 33:337-351. doi:10.1016/0033-5894(90)90060-X

---

## Referee Comment (RC3) · Anonymous Referee #3 · 15 Dec 2018

Major comments:

This article has a potential to be a valuable scientist contribution but it needs, in my opinion, much additional work prior to publication. There are a number of issues that I would like to see addressed, or at least discussed, by the authors. In order to make them easier to follow, I list them in bullet points below (in no particular order of importance):

* A discussion is needed with regard to what extent any, likely non-linear, temperature sensitivity in the data affects the precipitation reconstruction. Given the region, some temperature influence on the reconstructed precipitation signal is likely to exists and such a signal is likely to be non-linear and thus different between warmer and colder climate states during the past two millennia.

[Figure]

* I would like to see some "sensitivity tests" in the use of calibration window by using other seasonal windows than February 26 to July 6. It would be interesting to see how sensitive the skill of the reconstruction is to particular seasonal windows, especially as the presently used seasonal window is extremely well-defined down to single dates rather than months.

* I would like to see a longer discussion about the implication is the huge differences in MSL (see, e.g., Fig 2b). In my opinion, this is likely to result in a bias of the results to a larger extent than the authors acknowledge. Even if the problem cannot be solved (although it may be possible to use a subset of the data of the same segment length to conduct a "sensitivity test"), it needs to be discussed much more critically.

* It would be better to present all the result with regard to the climate mean of 1961–1990 instead of the mean of 1901–2000. This would make the results more comparable to other studies.

* The implications of the detrending choice much critically be discussed and the possibilities, or limitations, to apply RCS detrending (or "signal-free" detrending) must be seriously addressed.

Minor comments:

Page, 1, lines 13–15: This first sentence of the Abstract seems a bit out of place as the article only addresses the past two millennia and not the whole Holocene.

Page, 1, lines 27–28: The exact amplitude of the precipitation reduction is likely very sensitive to scaling/regression method.

Page 2, lines 1–3: The introduction is a bit vague and a bit out of place here. It is simply too general and not clearly related to the research problem in the article.

Page 2, lines 6–8: This is actually wrong. A number of two millennium-long calibrated precipitation reconstructions do exist. This is especially true for western North America. Moreover, MORE millennium-long hydroclimate reconstructions from tree-rings exist to

date than millennium-long tree-ring based reconstructions of temperature. This whole part needs to be rewritten and up to date with the present state of research.

Page 2, line 8: I would also cite here:

Cook, E.R., Woodhouse, C.A., Eakin, M., Meko, D.M., Stahle, D.W., 2004. Long-term aridity changes in the western United States. Science 306, 1015–1018.

Ljungqvist, F.C., Krusic, P.J., Sundqvist, H.S., Zorita, E., Brattström, G., Frank, D., 2016. Northern Hemisphere hydroclimate variability over the past twelve centuries. Nature 532, 94–98.

Prokop, O., Kolář, T., Büntgen, U., Kyncl, J., Kyncl, T., BošeÄ¿a, M., Choma, M., Barta, P., Rybníček, M., On the palaeoclimatic potential of a millennium-long oak ring width chronology from Slovakia. Dendrochronologia 2016, 40, 93–101.

Page 2, line 13: I would also cite:

Büntgen, U., Trouet, V., Frank, D., Leuschner, H.H., Friedrichs, D., Luterbacher, J., Esper, J., 2010. Tree-ring indicators of German summer drought over the last millennium. Quat. Sci. Rev. 29, 1005–1016. https://doi.org/10.1016/j.quascirev.2010.01.003.

Helama, S., Meriläinen, J., Tuomenvirta, H., Multicentennial megadrought in northern Europe coincided with a global El Niño–Southern Oscillation drought pattern during the Medieval Climate Anomaly. Geology 2009, 37, 175–178.

Klippel, L., Krusic, P. J., Brandes, R., Hartl, C., Belmecheri, S., Dienst, M., Esper, J., A 1286‐year hydro‐climate reconstruction for the Balkan Peninsula. Boreas 47 2018, 1218–1229.

Kress, A., Hangartner, S., Bugmann, H., Büntgen, U., Frank, D.C., Leuenberger, M., Siegwolf, R.T.W., Saurer, M., 2014. Swiss tree-rings reveal warm and wet summers during medieval times. Geophys. Res. Lett. 41, 1732–1737. http://dx.doi.org/10.1002/2013GL059081

Page 2, lines 16–18: Strange formulation here. What is said is unclear to me.

Page 4, Fig. 1: The Figure can be much improved, i.e. using ArcGIS or similar software, as well as be in colour for better clarity.

Page 5, Section 2.2: Are any references available for the various instrumental datasets from the various stations?

Page 5, line 22: Please, provide the standard references for RBAR and EPS.

Page 8, line 5: It should be mentioned that the Old World Drought Atlas is calibrated to scPDSI.

Page 8, lines 8–9: Are there any references to these datasets?

Page 8, line 10: For the Old World Drought Atlas (Cook et al., 2015), all included datasets are listed in the Supplement to the article in a table there.

Page 9, lines 7–9: Please, discuss the implication of these RBAR values more in detail.

Page 9, lines 12–13: A more detailed discussion about this problem is needed here.

Page 10, lines 11–12: A number of other references could also be added here.

Page 13, lines 17, 21: The word "connection" here could be replaced with "agreement" or "correlation".

Page 16, line 28: The same problem has also extensively been discussed in numerous other studies, e.g.:

Bürger, G., and U. Cubasch (2005), Are multiproxy climate reconstructions robust?, Geophys. Res. Lett., 32, L23711, doi:10.1029/2005GL024155.

Bürger, G., I. Fast, and U. Cubasch (2006), Climate reconstruction by regression—32 variations on a theme, Tellus A, 58, 227–235.

Christiansen, B. (2011), Reconstructing the NH mean temperature: Can underestimation of trends and variability be avoided?, J. Clim., 24, 674–692.

Christiansen, B. and Ljungqvist, F. C. 2017: Challenges and perspectives for large-scale temperature reconstructions of the past two millennia, Reviews of Geophysics, 55, 40–96.

Smerdon, J. E., A. Kaplan, D. Chang, and M. N. Evans (2011), A pseudoproxy evaluation of the CCA and RegEM methods for reconstructing climate fields of the last millennium, J. Clim., 24, 1284–1309.

Wang, J., J. Emile-Geay, D. Guillot, J. E. Smerdon, and B. Rajaratnam (2014), Evaluating climate field reconstruction techniques using improved emulations of real-world conditions, Clim. Past, 10, 1–19.

Page 17, line 4: Would blue intensity be an alternative to traditional density measurements in this context?

Page 19: Would not storage at the ITRDB also be a good option for long-term availability?

Page 27: It would be informative to also have a table for the wettest and driest decades.

Page 27: "Low pluvials" appears a strange expression to me. Do the authors means "Droughts" here?

---

## Author Comment (AC2) · 21 Dec 2018

Comments to Referee #2 by Alexander Land

Dear Referee #2, Thank you for these critical comments.

RC#2: Review of previous studies/reconstructions In a previous version of the manuscript we deeply reviewed other accessible reconstructions from Europe as well as from southern Germany. For example: Wilson et al. 2013 (DOI 10.1007/s00382-012-1318-z), Cooper et al. 2013 (DOI 10.1007/s00382-012-1328-x) or Wilson et al. 2005 (DOI 10.1002/joc.1150) to place much more stress on spatio-temporal differences between them and how important it is to get a long (two millennia) highly-resolved precipitation reconstruction entirely developed from a small region (here the Main region,

southern Germany). But this was criticised by a colleague, which caused us to use reconstructions only from the same grid box (here from Pauling et al. 2006, Cook et al. 2015) and for Central Europe (Büntgen et al. 2011).

RC#2: Mechanism and origin of the precipitation variability (e.g. influence of NAO) To my best knowledge the NAO has a strong influence during the winter period on the weather in Central Europe. Thus I would assume that there should be no significant influence. As can be read in Qian et al. (2000, https://doi.org/10.1029/2000JD900102): [...The North Atlantic Oscillation plays an important role in nonseasonal variability over the sector and leaves a significant signature in precipitation. But it does not seem to be the most important signal of atmospheric variability in precipitation over Europe, although it does in winter.]. But, to be honest, I did not analyse it, but I will.

RC#2: TRW chronology does not track extremely low/high precipitation rates This is a crucial point. In most of the tree-ring studies dealing with oaks this fact is present in the data. But only in a few of them it is explicitly mentioned/shown that extreme low rainfall is only poorly modelled by oak ring-width data (e.g. see Wilson et al. 2013 (Fig. 7, DOI 10.1007/s00382-012-1318-z), Copper et al. 2013 (Fig. 10, DOI 10.1007/s00382-012-1328-x)). In my opinion this point needs more attention in general and thus it is explicitly mentioned in our work.

---

## Author Comment (AC3) · 21 Dec 2018

Comments to Referee #3 by Alexander Land

Dear Referee #3, Thank you for these critical comments.

RC#3: Sensitivity tests using other seasonal windows than the well-defined (Feb. 26-Jul. 06) These sensitivity tests were conducted at the beginning of the project. The temporal changes as well as the changes in the length of the sensitive interval were studied including the well-defined interval Feb. 26-Jul. 06 as well as "classical" seasonal window lengths (e.g. April-June, March-July). This can be provided as a Figure in the Appendices. However, the "classical" seasonal windows are always well-defined, too: e.g. April-June = April 01-June 30. So to me our well-defined interval is nothing

else than a prolonged "classical" window.

RC#3: Longer discussion about the implication of differences in MSL This issue will be addressed and critically discussed in the associated sections. Here I would like to mention, that the minimum MSL is 110 and the maximum is 230 years. In the Figure 2b the fluctuation in MSL seems to be huge, because of the scale (y-axis) ranging from 90-240. The range of this axis was explicitly used to give the reader a more detailed information about the fluctuation of MSL. The fluctuation of MSL is not as "huge" as the line graph implies. Compared to other oak tree-ring studies, in my opinion, it is not really dramatic (even normal when dealing with subfossil and historical tree-ring series). Are there reasons for this assumption? Would it be possible to provide a reference to that topic?

RC#3: Climate mean 1961-1990 instead of 1901-2000 Mean precipitation sum (Feb. 26-Jul. 06) in the investigated Main region is 219.7 mm during the past century (1901-2000), whereas the mean precipitation sum from 1961 to1990 is 241.0 mm (+10%). The reason for choosing the one-century reference period has the following reason: "outside" 1961-1990 some more droughts appeared (e.g. 1921, 1934, 1991, 1993) characterising the climate of the region. When the previous millennia are compared to the reference period 1901-2000, in my opinion, it becomes much clearer that in some periods the droughts/wets are more "outstanding". In many other studies the reference period 1901-2000 is also used and sometimes the reference period 1971-2000 can be found (e.g. Karl et al. 2009: Global Climate Change Impacts in the United States or Meehl et al. 2003: Solar and greenhouse gas forcing and climate response in the twentieth century. Journal of Climate, 16, 426-444, DOI:10.1175/1520-0442(2003)0162.0.co). But perhaps I am wrong and we should change it to make it more comparable with other results.

RC#3: Critical discussion about applying RCS detrending – possibilities, limitations The 100-year MSL ($\sim$ minimum) restricts the potential to get low frequency information from such data, and one can use a 100-year smoothing spline. But indeed, our

discussion about that topic is not seriously addressed. In the revised version of the manuscript this will be critically discussed.

---

## Author Response (AR1)

[revised manuscript text omitted]

**Referee #1**

RC#1: Dear authors, I very much appreciate the attempt of analyzing and publishing some (why not all) of the unique oak ring-width data stored at the University of Hohenheim, Germany. This is a very important step towards generating exciting new science.

RC#1: However, I am reluctant in recommending acceptance of the submitted work, mainly because it represents an intermediate step rather than drawing methodological-sound conclusions from a final dataset (i.e. entire Holocene). This stepwise publishing procedure seems unnecessary in the case of central European oak ring-width measurements, as it has been successfully demonstrated in the recent past that such data are useful (and most relevant indeed) for reconstructing hydroclimate (i.e. a more complex reflection of spring to early-summer soil moisture availability instead of simple precipitation totals) on inter-annual to multi-centennial time-scales. While this has been done for several regions in Germany, France and England, Ed Cook's OWDA describes a European-wide milestone with regard to spatially explicit reconstructions of droughts and pluvials during the Common Era.

AC: I think there is a misunderstanding here. Ring-width data stored at the University of Hohenheim (Germany) are regularly contributed to different projects/studies to generate exciting science. Tree-ring data from the dendrolab Hohenheim are contributed to a wide variety of different dendro projects (only a few examples are given): 1) "Five millennia of European hydroclimate" (head of the project: W. Tegel, University of Freiburg, Germany; U. Büntgen, University of Cambridge, UK), 2) "Long-term trends in European tree growth over the past 1000 years - an interspecies comparison" (head of the project: A. Seim, University of Freiburg, Germany, funded by the DFG, project no. 389131207). Most of the Hohenheim ring-width data are already provided via data repositories (e.g. Zenodo or PANGAEA). Smaller data(sub)sets are indeed not published yet and are only accessible via personal contact/correspondence. This is mainly ring-width data from very local findings or sites (e.g. from the Rhine river) spanning a few centuries within the Holocene, but NOT the entire Holocene. For the presented study, tree-ring series (you called that a data(sub)set) from a specific region (here the Main region, southern Germany) were needed to develop a regional hydroclimate record and to compare this record to already existing reconstructions (during the past two millennia). The original (raw) dataset used here is made accessible to give others the opportunity to specifically reproduce our results and to have unrestricted access to all data underlying our study. This is transparent, in line with the data policy of the Copernicus Publications and good practical science. This is not in any way "politically" motivated or "strategically" aligned. Any suspicions that we have deliberately withheld tree-ring data are completely unfounded. We, as a research group, have strived in recent years to make our institution a positive example of transparency and scientific cooperation. We are more than happy to collaborate with and provide our data to others in the hopes of gaining new insights into past climate.

RC#1: In addition, the authors suggest that some of their data has been already incorporated into earlier studies (Büntgen et al., 2011, Cook et al., 2015), therefore only limited comparisons with these reconstructions are possible. Why did not the authors clarify this before? I am confident Büntgen and Cook would provide this information to the authors.

AC: I am pretty sure, and so I do agree with your comment, that if requested, Büntgen and Cook would have provided their datasets for a detailed analysis regarding duplicates in the dataset used here, which would allow for a clear statement of independence/dependence between the different reconstructions. Perhaps I am totally wrong and the original datasets are accessible via a data repository, or there were good reasons not to make these datasets accessible/public for unknown reasons. The widely accepted data policy of scientific journals

requires all authors to make materials, data etc. available. So one could argue that an independent study should be feasible (even without a direct correspondence during the publication process). I would like to mention that the mistake in this regard could be that I have simply failed to find the data (and the original datasets are accessible/public), meaning that this part of the manuscript has to be reanalyzed and modified.

RC#1: In short, the submitted work does not provide any ground-breaking methodological and/or intellectual novelty, and the relatively small data(sub)set does not appear to be robust between 800 and 1100 C.E. and again during the 4th century C.E. when the sample size dramatically drops (see Fig. 2 of the submitted draft). Although the EPS is above the common applied threshold of 0.85, the temporal replication changes can strongly affect the chronology behavior. Possible uncertainties might emerge from the integration of predominantly juvenile or mature/adult wood during these periods. Moreover, it is a pity that the low-frequency hydroclimate variability is not expressed in the presented reconstruction.

AC: In this study we used precipitation records with a daily resolution, which is relatively new to dendroclimatology (chapter 2.4 Calibration, verification and reconstruction of hydroclimate variability). The applied bootstrapped transfer function stability (BTFS) test to assess the temporal stability of the relationship between ring-widths and daily precipitation data (first introduced by Buras et al. 2017) is another example for a new (and innovative) method that was used. While I do agree that most of the presented results were obtained using standard and widely accepted dendroclimatological methods, the presented results nevertheless show e.g. that in the first millennium C.E. (fully-independent dataset) differences to other reconstructions (B11, C15) appear, which could be due to local/regional precipitation characteristics (see chapter 3.3 Comparison of MR reconstruction to others, Fig. 5, Fig. 6). This underlines the need to set up as much as possible local/regional hydroclimate reconstructions (even when standard methods are applied) to study spatial and temporal rainfall variability in the near future. Thus, this work does indeed provide additional information leading to a more detailed understanding of climate variability.
In the past few years intensive sampling of subfossil trees in the Main region was conducted, but did not lead to an increase in sample size in the 4th century and from 800-1100 C.E. The drop in sample size (as well as in the mean segment length) in the mentioned periods give evidence for fundamental environmental changes in the Main valley. It is possible that uncertainties in the reconstruction could emerge from the use of predominantly juvenile trees in these periods, but it underlines the statement (see previous comment) that there is a strong need to develop hydroclimate reconstructions on local/regional scale with tree-ring width datasets to ensure whether the uncertainty occurred from e.g. changing sample size.

**Referee #2:**

RC#2: Dear editor and authors of the manuscript "Two millennia of Main region (southern Germany) hydroclimate variability". To the best of my knowledge, the 2000 years long chronology is a novel idea through integration all available tree-ring width samples in this work, which would be an important contribution in the denderochronology community. Another new information is to calibrate the tree-ring width chronology using the daily instrumental data. However, the robust of the reconstruction should be furtherly analyzed, and the mechanism of precipitation variability should be conducted for reader to understand the origin of variability in the high-impart journal Climate of the past. Thus, I suggest that the manuscript should be accepted for publication after a revision.

RC#2: There are very long chronologies in Europe where is a hotspot in dendroclimatology. It is highly encouraged to carefully review the previous studies to place much more stress on innovation or difference of this study. The current motivation of this study is not very attractive to me. e.g. the first sentence in the abstract, the climate reconstruction covering the entire Holocene is important, but the TRW chronology in this study only covers the past two millennia.

AC: In a previous version of the manuscript we deeply reviewed other accessible reconstructions from Europe as well as from southern Germany. For example: Wilson et al. 2013 (DOI 10.1007/s00382-012-1318-z), Cooper et al. 2013 (DOI 10.1007/s00382-012-1328-x) or Wilson et al. 2005 (DOI 10.1002/joc.1150) to place much more stress on spatio-temporal differences between them and how important it is to get a long (two millennia) highly-resolved precipitation reconstruction entirely developed from a small region (here the Main region, southern Germany). But this was criticised by a colleague, which caused us to use reconstructions only from the same grid box (here from Pauling et al. 2006, Cook et al. 2015) and for Central Europe (Büntgen et al. 2011).

Author's changes: The first sentence in the abstract will be rephrased.

FINAL Author's changes: A reconstruction of hydroclimate with an annual or sub-annual resolution covering millennia for a geographically limited region in continental Europe would significantly improve our knowledge of past climate dynamics.

RC#2: The mechanism and origin of the precipitation variability (e.g. the influence of the Northern Atlantic Oscillation) should to be furtherly analyzed through comparison of the other reconstructions or model simulation. Another option is to select a more specific journal, e.g. dendrochronologia or Tree-Ring Research. The readers of Climate of the past would like to know some information about the mechanism of climate variability not only the phenomena.

AC: To my best knowledge the NAO, SO, AMO ENSO etc. have no significant influence on long-term rainfall variability in Central Europe. Thus I would assume that there should be no significant influence also on our dataset. As can be read in Qian et al. (2000, https://doi.org/10.1029/2000JD900102): [...The North Atlantic Oscillation plays an important role in nonseasonal variability over the sector and leaves a significant signature in precipitation. But it does not seem to be the most important signal of atmospheric variability in precipitation over Europe, although it does in winter.]. Brázdil et al. (2015, https://doi.org/10.1002/joc.4065) studied the forcings of spring-summer droughts in the Czech Land and found that [...solar irradiance and Southern Oscillation (SO) made only minor contributions to central European drought variability, while the effect of ... Atlantic Multidecadal Oscillation (AMO) were weaker and statistically insignificant.]. But on the other hand a study performed by Miksovsky et al. (Clim. Past Discuss., https://doi.org/10.5194/cp-2018-61) might give some hints that long-term drought variability could be forced by AMO.

Author's changes: Actually we intensively compare/analyse the influence of reconstructed AMO from Gray et al. (2004, doi: 10.1029/2004GL019932), Mann et al. (2009, doi: 10.1126/science.1177303), Singh (2018, https://doi.org/10.5194/cp-14-157-2018), sunspot number etc. on our developed spring-summer precipitation time series via cross-wavelet analysis and other suitable methods. In the revised version these results could be part of the discussion or part of a new section. Currently no results/Figures can be provided. But we work on this topic very intensively together with some other specialists.

FINAL Author's changes: After intensively analyses of a possible (long-term) mechanism /origin of the reconstructed precipitation variability in the Main region (e.g. via cross-wavelet analyses, XWTs), we conclude that the influence of large-scale circulation patterns (e.g. Atlantic Multidecadal Oscillation (AMO), North Atlantic Oscillation (NAO), Artic Oscillation (AO), Sea Level Pressure (SLP)) remains unclear. Please find attached a Figure with results from two XWTs. This is just a small sub-set of the conducted analyses / calculations. At first glance, these "promising" results indicate that the AMO (here two reconstructed AMO time series were used (Gray et al. 2004 / Mann et al. 2009)) may trigger the rainfall variability at the study area (Main region) on a long-term perspective (~16-20 and ~30-60 year frequency band). BUT when the phases (arrows in the Figure) are considered, they shift from phase (arrows pointing to the right) to anti-phase (arrows pointing to the left). Especially on the shorter frequency bands the results indicate that the relationship between AMO (as well as NAO, AO etc.) is not as clear as these XWTs (and our other performed analyses) suggest. Other studies (see Küttel et al. 2011, DOI: 10.1007/s00382-009-0737-y or Jacobeit et al. 2003, DOI: 10.1007/s00382-002-0278-0 and references therein) showed the pronounced complexity of the large-scale atmospheric circulation and European climate (especially for central Europe!). In other regions, like western North America or for parts of Asia, the connection between atmospheric circulation patterns and "regional / local" weather / climate is, to some extend, closer related.
So, in our opinion this phenomenon needs to be generally investigated much more intense for that specific study area from a meteorological point of view. We are aware that this is very unsatisfactory from a scientific point of view, but an intense investigation about the relationship of these large-scale circulations on the weather / climate in the study area, however, would blow up the content of this manuscript. We would therefore recommend not to include all these results / analyses, as this would make the clear presentation in the actual manuscript lost.

RC#2: The logic of the article is a bit problematic. The main target in this study is to reconstruct the precipitation variability over the past millennia. However, the following some evidences and discussion do not support this reconstruction. e.g. Page 10, line 9 'The TRW chronology does not track extremely low or high precipitation rates adequately.' The sections 4.3 also shows this weak relationship between the TRW chronology and the instrumental precipitation. Even the authors emphasize that the human influence may have a severe impact on forest in Page 17, lines 12-20. All prove that it is not very reasonable to reflect the extreme precipitation events over the past millennia. Another option to try to reconstruct the 'mixed' variable, e.g. the PDSI index.

AC: This is a crucial point. In most of the tree-ring studies dealing with oaks this fact is present in the data. But only in a few of them it is explicitly mentioned/shown that extreme low rainfall is only poorly modelled by oak ring-width data (e.g. see Wilson et al. 2013 (Fig. 7, DOI 10.1007/s00382-012-1318-z), Copper et al. 2013 (Fig. 10, DOI 10.1007/s00382-012-1328-x) and many, many others). In my opinion this point needs more attention in general and thus it is explicitly mentioned in our work. So, the logic of the manuscript is not problematic.

RC#2: Specific Comments: 1. Page 1, Lines 18-19. The bootstrap method is not an innovative analysis in dendroclimatology, please see the literatures e.g. (Guiot 1991; Till and Guiot 1990).

Author's changes: The sentence will be rephrased: To test the stability of the developed transfer function a bootstrapped transfer function stability test (BTFS) as well as …

FINAL Author's changes: Done

RC#2: Page 2, Line 3. It is difficult to predict future impact through climate reconstruction. The climate model is usually used to project the future scenario.

AC: As also mentioned by Referee #3 the Introduction needs some rewording and will be rephrased.

FINAL Author's changes: See changes in the chapter Introduction.

RC#2: Page 6, Line 13. Why is the 100-year cubic smoothing spline used to detrend the tree growth? To my knowledge, the standard standardization method depends on the special situation of each sample.

AC: The 100-year cubic smoothing spline was used to preserve as much as high- to mid-frequency as possible. Due to the minimum in the MSL of ~100 years, the length of the chosen smoothing spline is appropriate. Using the standard chronology does not severely impact how much decadal and longer term information is extracted from these data.

RC#2: Page 8. The section 2.5 should be moved to the introduction to emphasize the innovation of this paper through a review of previous reconstructions.

Author's changes: Parts of the section 2.5 will be moved to the Introduction.

FINAL Author's changes: Due to the required changes in the chapter Introduction, this point is taken up. Please see changes in the text.

RC#2: Page 8, Line 26. The reason selecting 51-year should be given. As we known, the window size would affect the results of running correlation.

AC: The relatively "short" 51-year window length was chosen to show agreements between the time series even on short time scale. At the beginning of the project a 101-year running window was also used, but the results remain more or less the same.

Author's changes: A sentence will be added to state why a 51-year window was used.

FINAL Author's changes: The following sentence was added: This relatively short window length allows us to study abrupt temporal changes in the behavior of the aforementioned reconstructions.

RC#2: The discussion of phase variability in the cross-wavelet transform and squared wavelet coherence is ignored. e.g. The MR and B11 has an obvious variability in phase in the upper right panel of Figure 5.

Author's changes: A section about phase variability will be added in the discussion part.

FINAL Author's changes: Due to the importance of the coherence analyses a section in chapter 3.3 was added: Coherence between MR-B11 and MR-C15 is also obvious prior to AD 1000, on short as well as on long terms, but not as high as in the second millennium. The right-pointing arrows in the significant regions (thick contours in the right panels of Fig. 5) indicate that the compared reconstructions clearly swing in phase, holding evidence for close time-frequency connections. The previously mentioned weak connection between MR and C15 in some periods (Fig. A1) becomes more obvious from the WTC. From AD 200–400, for example, no significant connection on shorter and even on longer time scales can be found, supporting the results from the running correlation analysis and indicating substantial differences between the different reconstruction series to the regional MR. The weaker, or even non-significant, connections between MR-B11 and MR-C15 in the first millennium AD show that there is a particularly low coherence between the regional hydroclimate reconstruction (MR) and B11 / C11 when an independent data set (see chapter 2.5) is used.

In the discussion section (see first part of chapter 4.3), this topic is taken up again. Please see changes in the text.

RC#2: Page 18, Line 24. The seasonal resolution would lead a misunderstanding. Here, it is really an annual resolution.

Author's changes: Will be changed in annual resolution.

FINAL Author's changes: Done

**Referee #3:**

RC#3: This article has a potential to be a valuable scientist contribution but it needs, in my opinion, much additional work prior to publication. There are a number of issues that I would like to see addressed, or at least discussed, by the authors. In order to make them easier to follow, I list them in bullet points below (in no particular order of importance):

RC#3: A discussion is needed with regard to what extent any, likely non-linear, temperature sensitivity in the data affects the precipitation reconstruction. Given the region, some temperature influence on the reconstructed precipitation signal is likely to exists and such a signal is likely to be non-linear and thus different between warmer and colder climate states during the past two millennia.

Author's changes: A section within the discussion will be added.

FINAL Author's changes: A section was added (4.2): While the oak tree-ring series used here respond very well to spring-summer rainfall during the calibration / verification period, it is not unlikely that, to some extent, warmer / colder phases during the past two millennia (e.g. the Medieval Climate Anomaly or the Little Ice Age) affect the presented reconstruction. It has been shown by Friedrichs et al. (2008) that oak trees from central-west Germany (which is close to our study region) lose their precipitation sensitivity in the anomalously warm decade 1940s. This decreased response to hydroclimatic conditions in central Germany has been confirmed by Büntgen et al. (2010a), whose study revealed that oak TRW sensitivity is greatly reduced to scPDSI in the mid-20th century. A well-established decrease in precipitation sensitivity during the 1940s is also observed in our study (data not shown). With this in mind, it is possible that during extraordinary warm (Medieval Climate Anomaly) or extraordinary cold (Little Ice Age) (Mann et al., 2009) periods, TRW reconstructions may show a certain level of bias. However, it remains unclear to what extent the reconstruction here presented is biased.

RC#3: I would like to see some "sensitivity tests" in the use of calibration window by using other seasonal windows than February 26 to July 6. It would be interesting to see how sensitive the skill of the reconstruction is to particular seasonal windows, especially as the presently used seasonal window is extremely well-defined down to single dates rather than months.

AC: These sensitivity tests were conducted at the beginning of the project. The temporal changes as well as the changes in the length of the sensitive interval were studied including the well-defined interval Feb. 26-Jul. 06 as well as "classical" seasonal window lengths (e.g. April-June, March-July). This can be provided as a Figure/Table (or in the text) in the Appendices. However, the "classical" seasonal windows are always well-defined, too: e.g. April-June = April 01-June 30. So to me our well-defined interval is nothing else than a prolonged "classical" window.

FINAL Author's changes: A new Table (Table A1) was added in the Appendices and the following sentence was added in section 3.2: For sensitivity test results of the TRW chronology with total precipitation sum for "classical" monthly-resolved seasons, see Table A1.

RC#3: I would like to see a longer discussion about the implication is the huge differences in MSL (see, e.g., Fig 2b). In my opinion, this is likely to result in a bias of the results to a larger extent than the authors acknowledge. Even if the problem cannot be solved (although it may

be possible to use a subset of the data of the same segment length to conduct a "sensitivity test"), it needs to be discussed much more critically.

AC: Here I would like to mention, that the minimum MSL is 110 and the maximum is 230 years. In the Figure 2b the fluctuation in MSL seems to be huge, because of the scale (y-axis) ranging from 90-240. The range of this axis was explicitly used to give the reader a more detailed information about the fluctuation of MSL. The fluctuation of MSL is not as "huge" as the line graph implies. Compared to other oak tree-ring studies, in my opinion, it is not really dramatic (even normal when dealing with subfossil and historical tree-ring series).

Author's changes: However, this issue will be addressed and critically discussed in the associated sections.

FINAL Author's changes: This section (3.1) was rephrased to address your comment. For detailed changes please see text.

RC#3: It would be better to present all the result with regard to the climate mean of 1961–1990 instead of the mean of 1901–2000. This would make the results more comparable to other studies.

AC: Mean precipitation sum (Feb. 26-Jul. 06) in the investigated Main region is 219.7 mm during the past century (1901-2000), whereas the mean precipitation sum from 1961 to1990 is 241.0 mm. The reason for choosing the one-century reference period has the following reason: "outside" 1961-1990 some more droughts appeared (e.g. 1921, 1934, 1991, 1993) characterising the climate of the region. When the previous millennia are compared to the reference period 1901-2000, in my opinion, it becomes much clearer that in some periods the droughts/wets are more "outstanding". In many other studies the reference period 1901-2000 is also used (and sometimes 1971-2000 can be found), see also e.g. Karl et al. 2009: Global Climate Change Impacts in the United States or Meehl et al. 2003: Solar and greenhouse gas forcing and climate response in the twentieth century. Journal of Climate, 16, 426-444, DOI:10.1175/1520-0442(2003)0162.0.co.

RC#3: The implications of the detrending choice much critically be discussed and the possibilities, or limitations, to apply RCS detrending (or "signal-free" detrending) must be seriously addressed.

AC: The 100-year MSL (~ minimum) restricts the potential to get low frequency information from such data, and one can use a 100-year smoothing spline. But indeed, our discussion about that topic is not seriously addressed.

Author's changes: In the revised version of the manuscript this point will be seriously addressed/critically discussed.

FINAL Author's changes: The following passage was added to the discussion section 4.2: Nevertheless, due to the changes in the mean segment length and the highly variable sample replication, the standardization procedure applied here (100-year spline) is suitable for preserving high- to mid-frequency fluctuations, while the low-frequency variance from the TRW data set is removed. Thus no inferences can be made for centennial-long precipitation fluctuations for the study region. When comparing the MR hydroclimate reconstruction to B11 and C15 (both capturing the low-frequency domain e.g. by using RCS detrending), some differences appear on the low-frequency timescale. These are especially apparent in the first

millennium AD, where trees from alluvial deposits are available for reconstruction purposes, and is more pronounced between MR-B11 than between MR-C15.

RC#3: Page, 1, lines 13–15: This first sentence of the Abstract seems a bit out of place as the article only addresses the past two millennia and not the whole Holocene.

Author's changes: The sentence will be deleted.

FINAL Author's changes: Due to required changes of Reviewer #2, this sentence was rephrased: A reconstruction of hydroclimate with an annual or sub-annual resolution covering millennia for a geographically limited region in continental Europe would significantly improve our knowledge of past climate dynamics.

RC#3: Page, 1, lines 27–28: The exact amplitude of the precipitation reduction is likely very sensitive to scaling/regression method.

AC: Please specify this comment.

RC#3: Page 2, lines 1–3: The introduction is a bit vague and a bit out of place here. It is simply too general and not clearly related to the research problem in the article.

Author's changes: The introduction will be rephrased to clearly relate to the research problem.

FINAL Author's changes: The Introduction was revised to make it more clear and concise. Please see our changes in the text.

RC#3: Page 2, lines 6–8: This is actually wrong. A number of two millennium-long calibrated precipitation reconstructions do exist. This is especially true for western North America. Moreover, MORE millennium-long hydroclimate reconstructions from tree-rings exist to date than millennium-long tree-ring based reconstructions of temperature. This whole part needs to be rewritten and up to date with the present state of research.

Author's changes: This part will be rewritten.

FINAL Author's changes: This part has been specified. Please see changes in the text of the Introduction.

RC#3: Page 2, line 8: I would also cite here:
Cook, E.R., Woodhouse, C.A., Eakin, M., Meko, D.M., Stahle, D.W., 2004. Long-term aridity changes in the western United States. Science 306, 1015–1018.
Ljungqvist, F.C., Krusic, P.J., Sundqvist, H.S., Zorita, E., Brattström, G., Frank, D., 2016. Northern Hemisphere hydroclimate variability over the past twelve centuries. Nature 532, 94–98.

Prokop, O., Kolá˘r, T., Büntgen, U., Kyncl, J., Kyncl, T., BošeÄ¿a, M., Choma, M., Barta, P., Rybní˘cek, M., On the palaeoclimatic potential of a millennium-long oak ring width chronology from Slovakia. Dendrochronologia 2016, 40, 93–101.

Author's changes: Cook et al. 2004 and Ljungqvist et al. 2016 will be cited.

FINAL Author's changes: Cook et al. 2004, Ljungqvist et al. 2016 and Prokop et al. 2016 was cited.

RC#3: Page 2, line 13: I would also cite:
Büntgen, U., Trouet, V., Frank, D., Leuschner, H.H., Friedrichs, D., Luterbacher, J., Esper, J., 2010. Tree-ring indicators of German summer drought over the last millennium. Quat. Sci. Rev. 29, 1005–1016. https://doi.org/10.1016/j.quascirev.2010.01.003.
Helama, S., Meriläinen, J., Tuomenvirta, H., Multicentennial megadrought in northern Europe coincided with a global El Niño–Southern Oscillation drought pattern during the Medieval Climate Anomaly. Geology 2009, 37, 175–178.
Klippel, L., Krusic, P. J., Brandes, R., Hartl, C., Belmecheri, S., Dienst, M., Esper, J., A 1286â˘A ˘ Ryear hydroâ˘AR˘ climate reconstruction for the Balkan Peninsula. Boreas 47 2018, 1218–1229.
Kress, A., Hangartner, S., Bugmann, H., Büntgen, U., Frank, D.C., Leuenberger, M., Siegwolf, R.T.W., Saurer, M., 2014. Swiss tree-rings reveal warm and wet summers during medieval times. Geophys. Res. Lett. 41, 1732–1737.
http://dx.doi.org/10.1002/2013GL059081

Author's changes: Helama et al. 2009 and Kress et al. 2014 will be cited.

FINAL Author's changes: Done

RC#3: Page 2, lines 16–18: Strange formulation here. What is said is unclear to me.

Author's changes: The sentence will be rephrased.

FINAL Author's changes: The sentence was deleted.

RC#3: Page 4, Fig. 1: The Figure can be much improved, i.e. using ArcGIS or similar software, as well as be in colour for better clarity.

AC: In general, a black/white map is as good as a coloured one as long all relevant information can be gained. But I do agree, that some additional information (e.g. Long./Lat.) can be added.

Author's changes: A coloured Figure will be provided. Long./Lat. will be added (see attached Figure).

FINAL Author's changes: Done (see new Fig. 1 in the revised manuscript). The Figure caption has been adjusted accordingly.

RC#3: Page 5, Section 2.2: Are any references available for the various instrumental datasets from the various stations?

AC: The references are stated (line 7 on page 5).

RC#3: Page 5, line 22: Please, provide the standard references for RBAR and EPS.

Author's changes: Wigley et al. (1984). J Clim Appl Meteorol 23:201–213 will be provided.

FINAL Author's changes: Done

RC#3: Page 8, line 5: It should be mentioned that the Old World Drought Atlas is calibrated to scPDSI.

Author's changes: It will be mentioned.

FINAL Author's changes: Done

RC#3: Page 8, lines 8–9: Are there any references to these datasets?

AC: The references are mentioned in lines 2-6.

Author's changes: … from the above mentioned references… will be added.

FINAL Author's changes: Changed in: The mentioned reconstruction series…

RC#3: Page 8, line 10: For the Old World Drought Atlas (Cook et al., 2015), all included datasets are listed in the Supplement to the article in a table there.

AC: This is correct. All included datasets are listed.

Author's changes: The sentence will be rephrased: The set of original, single TRW series included in the respective reconstructions by the mentioned authors is, to the best of our knowledge, not explicitly stated in their work or accessible. This circumstance made a comparison between the respective records extremely difficult.

FINAL Author's changes: Done

RC#3: Page 9, lines 7–9: Please, discuss the implication of these RBAR values more in detail.

Author's changes: A more detailed discussion will be given.

FINAL Author's changes: This section (3.1) was rephrased to address your comment. For detailed changes please see text.

RC#3: Page 9, lines 12–13: A more detailed discussion about this problem is needed here.

Author's changes: A more detailed discussion will be given.

FINAL Author's changes: This section (3.1) was rephrased to address your comment. For detailed changes please see text.

RC#3: Page 10, lines 11–12: A number of other references could also be added here.

Author's changes: To make clear that the given references are only a small selection "e.g." will be added.

FINAL Author's changes: Done

RC#3: Page 13, lines 17, 21: The word "connection" here could be replaced with "agreement" or "correlation".

Author's changes: The word connection will be replaced by agreement.

FINAL Author's changes: Done

RC#3: Page 16, line 28: The same problem has also extensively been discussed in numerous other studies, e.g.:
Bürger, G., and U. Cubasch (2005), Are multiproxy climate reconstructions robust?, Geophys. Res. Lett., 32, L23711, doi:10.1029/2005GL024155.
Bürger, G., I. Fast, and U. Cubasch (2006), Climate reconstruction by regressionăŎAˇT32 variations on a theme, Tellus A, 58, 227–235.
Christiansen, B. (2011), Reconstructing the NH mean temperature: Can underestimation of trends and variability be avoided?, J. Clim., 24, 674–692.
Christiansen, B. and Ljungqvist, F. C. 2017: Challenges and perspectives for largescale temperature reconstructions of the past two millennia, Reviews of Geophysics, 55, 40–96.
Smerdon, J. E., A. Kaplan, D. Chang, and M. N. Evans (2011), A pseudoproxy evaluation of the CCA and RegEM methods for reconstructing climate fields of the last millennium, J. Clim., 24, 1284–1309.
Wang, J., J. Emile-Geay, D. Guillot, J. E. Smerdon, and B. Rajaratnam (2014), Evaluating climate field reconstruction techniques using improved emulations of real-world conditions, Clim. Past, 10, 1–19.

Author's changes: The following references will be added: Christiansen and Ljungqvist (2017) as well as Bürger and Cubasch (2005).

FINAL Author's changes: Done

RC#3: Page 17, line 4: Would blue intensity be an alternative to traditional density measurements in this context?

AC: To our best knowledge, traditional density and blue intensity (BI) measurements were only performed on coniferous wood. As our investigation deals with oak trees this would not be an alternative.

RC#3: Page 19: Would not storage at the ITRDB also be a good option for long-term availability?

AC: Storage at the chosen data repository (Zenodo) is a good option for long-term availability and due to the assigned doi number easy to access.

RC#3: Page 27: It would be informative to also have a table for the wettest and driest decades.

Author's changes: Table 1 will be extended. Wettest and driest decades will be added.

FINAL Author's changes: A new Table (Table A2) was added in the Appendices and a sentence was added to section 3.2: A detailed list of dry and wet decades is provided in Table A2.

RC#3: Page 27: "Low pluvials" appears a strange expression to me. Do the authors means "Droughts" here?

Author's changes: "Low pluvials" will be changed in "Droughts" and "High pluvials" will be changed in "Wets".

FINAL Author's changes: Pluvial was replaced by rainfall.

---

## Author Response (AR2)

Comments to Referee #3 by Alexander Land

Dear Editor, dear Referee #3,

Thank you again for these critical comments.

RC#3: I would like to see some additional discussion about why not RCS detrending have been attempted. It was applied – the success of it hard to deem – in Büntgen et al. (2011) on a rather similar material that was hardly less heterogeneous to its nature. Although the present study contributes with valuable new understanding at interannual to decadal scale of past spring to summer hydroclimate variability – especially with regards to extremes – the long-term trends that certainly existed would also have been valuable to capture. Given the short segment length of the material, only RCS detrending can achieve that.

AC: This comment has already been seriously addressed by the Author's (please see our changes in a previous version of the manuscript). If one compares the data set used here with other already published data sets, the reconstructions developed are similar on a decadal (to multi-decadal) scale, independent of the standardization method. This is particularly evident in Graph 5 in the manuscript, which shows that the use of a spline (used in our study) is a strong alternative to RCS detrending. The authors think that this comment has already been adequately addressed and critically discussed in the manuscript.

RC#3: I am delighted to see that the authors are going to make their data freely available. It will be very valuable for future research. However, it would also be good it the authors too despite the data at the International Tree-Ring Data Bank (ITRDB) as it will better safeguard long-term accessibility. Moreover, the actual calibrated reconstruction, with clear caveats regarding the above-mentioned limitations in long-term variability, should be made available at https://www.ncdc.noaa.gov/data-access/paleoclimatology-data

AC: In addition to the publication in the Zenodo research data repository, all data will additionally be stored at the International Tree-Ring Data Bank (ITRDB).

RC#3: Finally, the authors should consider to get their article carefully checked and edited for correct English language use. The present reviewer is not native speaker but it appears clearly that the English language in the article contains numerous errors in grammar, syntax, and general structure. Sometimes there is an unsuitable choice of word for the context too. The Reference list contains errors and mistakes and should be carefully copyedited.

AC: I like to mention that one of the authors is a native speaker (from the US) and another author lived in the US for more than 10 years. Most of the Authors have many years of experience in publishing. Our manuscript was meticulously language edited during the entire process. We are therefore sure that grammar, syntax and/ or general structure meets the high requirements of CP. The Reference has been carefully copyedited.

RC#3: Page 1, line 14: Remove the word "would".

AC: Done.

RC#3: Page 1, line 15: Remove the word "total".

AC: Done.

RC#3: Page 1, line 28: Add ~ before 38% and 39%.

AC: Done.

RC#3: Page 2, lines 9–11: Add reference: Cook, E.R., Woodhouse, C.A., Eakin, M., Meko, D.M., Stahle, D.W., 2004. Long-term aridity changes in the western United States. Science 306, 1015–1018. https://doi.org/10.1126/science.1102586 [Add to Citavi project by DOI] .

AC: This Reference is already included.

RC#3: Page 5, line 6: Any web link to where to obtain the DWD records?

AC: We do not consider this to be necessary as the access links are subject to constant change.

RC#3: Page 18, line 12: Would not a warm spring with moderate rainfall rather increase oak growth? If it is warm and enough soil moisture, I would have guessed that growth conditions would be rather favourable?

AC: I am not sure how to deal with that comment. The above mentioned statements are more or less in line with our findigs.

Comments to Referee #4 by Alexander Land

Dear Editor, dear Johannes Edvardsson (Referee #4),

Thank you for these critical comments.

RC#4: I find it remarkable that the word temperature is only mentioned twice (at line 5 and 6 in the introduction), as I expect temperature, not only precipitation, to be among the factors that influence the total ring width of oak trees in southern Germany. There is some discussion about influence during warm and cold conditions (line 4-10, page 18), but only with the conclusion that the presented reconstruction might be biased, but that it's unclear to what extent. In papers such as Cufar et al (2014, Trees 28, 1267-1277) they found negative spring-summer temperature signal in oak trees from central Europe. Another example of the importance of temperature might be the Little Ice Age (LIA), which has been considered as cold but dry period in many regions. Even though it was dry moisture sensitive trees in some regions suffered during LIA, maybe as a result of limited evaporation. I therefore suggest a presentation about to what extent both temperature and precipitation influence the tree growth in the result section and a broader discussion if the influence precipitation has on ring width changes during warm versus cold periods.

AC: A similar comment had been made by Reviewer #3 on a previous version of the manuscript, which has already been discussed extensively. Here our previous comment to Reviewer #3: A section was added (4.2): While the oak tree-ring series used here respond very well to spring-summer rainfall during the calibration / verification period, it is not unlikely that, to some extent, warmer / colder phases during the past two millennia (e.g. the Medieval Climate Anomaly or the Little Ice Age) affect the presented reconstruction. It has been shown by Friedrichs et al. (2008) that oak trees from central-west Germany (which is close to our study region) lose their precipitation sensitivity in the anomalously warm decade 1940s. This decreased response to hydroclimatic conditions in central Germany has been confirmed by Büntgen et al. (2010a), whose study revealed that oak TRW sensitivity is greatly reduced to scPDSI in the mid-20th century. A well-established decrease in precipitation sensitivity during the 1940s is also observed in our study (data not shown). With this in mind, it is possible that during extraordinary warm (Medieval Climate Anomaly) or extraordinary cold (Little Ice Age) (Mann et al., 2009) periods, TRW reconstructions may show a certain level of bias. However, it remains unclear to what extent the reconstruction here presented is biased.

RC#4: In the abstract, it's written that "a reconstruction of hydroclimate with annual to sub-annual resolution covering two millennia for continental Europe" will be presented. However, the word sub-annual is only mentioned once (line 5 in the introduction) in the entire manuscript. It therefore feels that more is promised in abstract than what is presented in the manuscript. Annual to decadal is in better agreement with the reconstructions presented in figure 6.

AC: Sub-annual was deleted.

RC#4: Please explain the arrows in the figure caption (Fig 5), so that figure and figure caption work independent from the text.

AC: Done.

[revised manuscript text omitted]
 rReconstruction of Ssummer pPrecipitation vVariability and Ddating of Fflood Eevents for the Mmillennium Bbetween 3250 and 2250 Yyears BC for the Main Region, Ssouthern Germany, in: Integrated Analysis of Interglacial Climate Dynamics (INTERDYNAMIC), Schulz, M., Paul, A. (Eds.), SpringerBriefs in Earth System Sciences, Springer International Publishing, Cham, 127–131, 2015.

Seftigen, K., Goosse, H., Klein, F., and Chen, D.: Hydroclimate variability in Scandinavia over the last millennium-insights from a climate model-proxy data comparison, Climate of the Past, 13, 1831–1850, doi:10.5194/cp-13-1831-2017, 2017.

Spurk, M., Leuschner, H. H., Baillie, M. G. L., Briffa, K. R., and Friedrich, M.: Depositional frequency of German subfossil oaks: Climatically and non-climatically induced fluctuations in the Holocene, The Holocene, 12, 707–715, doi:10.1191/0959683602hl583rp, 2002.

Steger, O.: Zur Eichenwicklerbekämpfung 1958 im Hochspessart, Forstwissenschafltiches Centralblatt, 78, 108–120, 1959.

Steger, O.: Spätfröste und Massenwechsel von *Tortrix viridiana* L. (Lep. Tortr.), Zeitschrift für angewandte Entomologie, 213–216, 1960.

Stocker, T. F., Qin, D., Plattner, G.-K., Tignor, M., Allen, S. K., Boschung, J., Nauels, A., Xia, Y., Bex, V., and Midgley P. M. (Eds.): IPCC 2013: Climate Change 2013: The Physical Basis. Contribution of Working Group I to the Fifth Assessment Report of the Intergovernmental Panel on Climate Change, Cambridge University Press, United Kingdom and New York, NY, USA, 2013.

Stockton, C. W. and Meko, D. M.: A long-term history of drought occurrence in western United States as inferred from tree rings, Weatherwise, 28, 244–249, 1975.

Storch, H. von, Zorita, E., Jones, J. M., Dimitriev, Y., González-Rouco, F., and Tett, Simon F. B.: Reconstructing pPast Cclimate from Nnoisy Ddata, Science, 306, 2004.

Wigley, T.M.L., Briffa, K. R., and Jones, P. D.: On the average value of correlated time series, with applications in dendroclimatology and hydrometeorology, Journal of Climate and Applied Meteorology, 23, 201–213, 1984.

Wilson, R., Anchukaitis, K., Briffa, K. R., Büntgen, U., Cook, E., D'Arrigo, R., Davi, N., Esper, J., Frank, D., Gunnarson, B., Hegerl, G., Helama, S., Klesse, S., Krusic, P. J., Linderholm, H. W., Myglan, V., Osborn, T. J., Rydval, M., Schneider, L., Schurer, A., Wiles, G., Zhang, P., and Zorita, E.: Last millennium northern hemisphere summer temperatures from tree rings: Part I: The long term context, Quaternary Science Reviews, 134, 1–18, doi:10.1016/j.quascirev.2015.12.005, 2016.

Wilson, R. and Elling, W.: Temporal instability in tree-growth/climate response in the Lower Bavarian Forest region: implications for dendroclimatic reconstruction, Trees - Structure and Function, 18, 19–28, doi:10.1007/s00468-003-0273-z, 2004.

Wilson, R., Miles, D., Loader, N. J., Melvin, T., Cunningham, L., Cooper, R., and Briffa, K.: A millennial long March-July precipitation recontruction for southern-central England, Climate Dynamics, 40, 997–1017, doi:10.1007/s00382-012-1318-z, 2013.

Wilson, R. J. S., Luckman, B. H., and Esper, J.: A 500 year dendroclimatic reconstruction of spring-summer precipitation from the lower Bavarian Forest region, Germany, International Journal of Climatology, 25, 611–630, doi:10.1002/joc.1150, 2005.

**Tables**

**Table 1. List of years with reconstructed far below / above rainfall (Feb. 26–Jul. 06) depicted as deviation (%) from the reference period AD 1901–2000.**

| Year AD | Low rainfall | Year AD | High rainfall |
|---------|--------------|---------|---------------|
| 338 | -38 | 357 | 39 |
| 337 | -37 | 985 | 37 |
| 1167 | -29 | 526 | 34 |
| 510 | -28 | 1533 | 34 |
| 1394 | -27 | 654 | 32 |
| 565 | -26 | 1317 | 29 |
| 945 | -26 | 1436 | 29 |
| 1165 | -26 | 460 | 29 |
| 1177 | -25 | 1314 | 28 |
| | | 436 | 28 |
| | | 559 | 28 |
| | | 1673 | 27 |
| | | 1123 | 27 |
| | | 1487 | 27 |
| | | 43 | 27 |
| | | 1052 | 26 |
| | | 1531 | 26 |
| | | 496 | 26 |
| | | 1055 | 26 |
| | | 56 | 26 |
| | | 778 | 26 |
| | | 1727 | 26 |
| | | 102 | 25 |
| | | 1056 | 25 |
| | | 932 | 25 |
| | | 602 | 25 |